# Vaccine Strategies Against RNA Viruses: Current Advances and Future Directions

**DOI:** 10.3390/vaccines12121345

**Published:** 2024-11-28

**Authors:** Kuei-Ching Hsiung, Huan-Jung Chiang, Sebastian Reinig, Shin-Ru Shih

**Affiliations:** 1Research Center for Emerging Viral Infections, College of Medicine, Chang Gung University, Taoyuan 33302, Taiwan; kchsiung@gap.cgu.edu.tw (K.-C.H.); chianghuanjung@gmail.com (H.-J.C.); plinigdererste@mail.de (S.R.); 2Graduate Institute of Biomedical Science, College of Medicine, Chang Gung University, Taoyuan 33302, Taiwan; 3Department of Laboratory Medicine, Linkou Chang Gung Memorial Hospital, Taoyuan 33305, Taiwan; 4Department of Medical Biotechnology & Laboratory Science, College of Medicine, Chang Gung University, Taoyuan 33302, Taiwan; 5Research Center for Chinese Herbal Medicine, Research Center for Food & Cosmetic Safety, Graduate Institute of Health Industry Technology, College of Human Ecology, Chang Gung University of Science & Technology, Taoyuan 33303, Taiwan

**Keywords:** RNA virus, vaccine platform, RNA virus vaccines, SARS-CoV-2, influenza virus, enterovirus, dengue virus, Zika virus, immune response

## Abstract

The development of vaccines against RNA viruses has undergone a rapid evolution in recent years, particularly driven by the COVID-19 pandemic. This review examines the key roles that RNA viruses, with their high mutation rates and zoonotic potential, play in fostering vaccine innovation. We also discuss both traditional and modern vaccine platforms and the impact of new technologies, such as artificial intelligence, on optimizing immunization strategies. This review evaluates various vaccine platforms, ranging from traditional approaches (inactivated and live-attenuated vaccines) to modern technologies (subunit vaccines, viral and bacterial vectors, nucleic acid vaccines such as mRNA and DNA, and phage-like particle vaccines). To illustrate these platforms’ practical applications, we present case studies of vaccines developed for RNA viruses such as SARS-CoV-2, influenza, Zika, and dengue. Additionally, we assess the role of artificial intelligence in predicting viral mutations and enhancing vaccine design. The case studies underscore the successful application of RNA-based vaccines, particularly in the fight against COVID-19, which has saved millions of lives. Current clinical trials for influenza, Zika, and dengue vaccines continue to show promise, highlighting the growing efficacy and adaptability of these platforms. Furthermore, artificial intelligence is driving improvements in vaccine candidate optimization and providing predictive models for viral evolution, enhancing our ability to respond to future outbreaks. Advances in vaccine technology, such as the success of mRNA vaccines against SARS-CoV-2, highlight the potential of nucleic acid platforms in combating RNA viruses. Ongoing trials for influenza, Zika, and dengue demonstrate platform adaptability, while artificial intelligence enhances vaccine design by predicting viral mutations. Integrating these innovations with the One Health approach, which unites human, animal, and environmental health, is essential for strengthening global preparedness against future RNA virus threats.

## 1. Introduction

RNA viruses have profoundly impacted both global health and regional and global economies; they account for 44% of all emerging contagious diseases [1]. Although all possess RNA as the genetic material, RNA viruses constitute a diverse group of pathogens that are very different in their biological characteristics and cause distinct clinical manifestations. Humans have long been in a constant battle against these life-threatening foes [2] since the first documented outbreak of measles [3]. The emergence and re-emergence of RNA viruses have led to widespread morbidity, mortality, and economic disruption. In the past century, highly pathogenic RNA viruses have caused devastating outbreaks, including the 1918 influenza pandemic, severe acute respiratory syndrome coronavirus 1 (SARS-CoV-1) in 2003, and severe acute respiratory syndrome coronavirus 2 (SARS-CoV-2) in 2019. Other significant outbreaks include the Zika virus epidemic from 2015 to 2016, dengue fever with its first report in 1635 [4], enterovirus outbreaks since 2014, and the Nipah virus outbreak in 2023. Despite advances in vaccine development, the onset of the COVID-19 pandemic in 2019, caused by SARS-CoV-2, is estimated to have resulted in over 7 million deaths and economic losses exceeding USD 10 trillion [5,6]. Similarly, even with the availability of seasonal influenza vaccines, influenza viruses continue to cause significant levels of morbidity and mortality each year [7]. Nonetheless, vaccination remains one of the most effective approaches to preventing viral infections and reducing viral transmission [8]. Vaccines are designed to stimulate the hosts’ immune responses against pathogens, thereby providing both individual and herd immunity [9]. Given the high genetic variability of RNA viruses, developing vaccines against these viruses and platforms for rapid and flexible vaccine production has become a major challenge. However, innovative developments in vaccine technology, including viral vectors, mRNA vaccines, and novel adjuvants, have demonstrated promise in addressing these difficulties. Compared with DNA viruses, RNA viruses have mutation rates that are hundreds of times higher and are thus characterized by more rapid rates of evolution and host adaptation, which often contribute to the emergence of epidemics or pandemics [10]. The emergence of new variants or viruses accordingly presents considerable challenges for effective vaccine development. However, therapeutic interventions or treatments for some virus-induced diseases remain costly or unavailable [11,12], underscoring the persistent unmet medical need for vaccination and the importance of future vaccine development. According to the World Health Organization, global vaccination programs have saved at least 150 million lives over the past 50 years, thereby highlighting the vital role vaccination plays in public health and the prevention of emerging viral diseases [13].

In this review, we provide a comprehensive overview of current strategies and advances in the development of vaccines against RNA viruses, along with the immune responses they elicit. We cover both traditional and modern vaccine platforms, highlighting their advantages and limitations, and present case studies of RNA virus vaccines based on the characteristics and epidemiological impact of major RNA viruses and the immune responses that these infections trigger. Additionally, we examine technological advances and emerging strategies, including universal and personalized vaccines and the One Health approach. By addressing these essential topics, we hope to inform researchers, healthcare professionals, and policymakers regarding the current state and future prospects of RNA virus vaccines, thereby contributing to global efforts to control and prevent RNA virus-associated diseases.

## 2. Immunity Induced by Viral Infections and Vaccinations

While many vaccine candidates are under investigation, the fundamental principle remains to safely elicit a robust immune response that mimics the protection achieved through natural infection. To this end, it is essential to explore both how body responds to RNA virus during natural infection and how vaccination can replicate these immune responses, providing long-lasting immunity without causing disease.

### 2.1. Innate Immune Response to Viral Infections

Innate immunity, which serves as the first line of defense against viruses, involves an immediate response that contributes to preventing infection. As primary mediators in this response, when stimulated by viral RNA, pattern recognition receptors, such as Toll-like receptors (TLRs) and retinoic acid-inducible gene I-like receptors, initiate downstream signaling pathways that lead to the production of type-1 interferons (IFNs) and inflammatory cytokines [14,15]. Type 1 IFNs play an important role in establishing an antiviral response by inducing the expression of interferon-simulated genes, including those involved in apoptosis, immune modulation, cell attraction and adhesion, and antiviral detection, which collectively contribute to an inhibition of viral replication and spread [16]. These responses are complemented by the activity of inflammatory cytokines, such as interleukins (ILs) and tumor-necrosis factors, that contribute to recruiting immune cells, including macrophages, natural killer cells, and dendritic cells, to the site of infection. Among these, natural killer cells can directly kill infected cells, whereas dendritic cells process and present viral antigens to activate the adaptive immune responses [17]. These innate immune responses are rapid and coordinated, and they play an essential role in preventing viral replication during the early stages of RNA virus infections and limiting viral dissemination [18,19]. However, RNA viruses have a number of counter-measures for suppressing host innate immune responses, employing complex strategies, such as interfering with antigen-presenting pathways, gene silencing, and protein cleavage [20,21]. For example, to evade apoptosis, the SARS-CoV-2 virus masks epitopes by camouflaging its spike protein with glycan molecules, inhibiting interferon production and disrupting complete mitophagy, thereby resulting in an increment in virus replication [22].

### 2.2. Adaptive Immune Response to Viral Infections

Adaptive immunity is characterized by an antigen-specific response against pathogens and a capacity to retain an immune memory. Among the different types of adaptive response, cellular adaptive immunity is conferred by T cells, whereas humoral immunity is promoted by antibodies produced by B cells. T cell immunity is the most important component of immunity against natural infections with RNA viruses, in which the action 8 of both CD4 T-helper cells and CD cytotoxic T-cells have been well-established as playing pivotal roles in protection against diseases caused by a diverse range of viruses, including influenza A virus, coronaviruses, respiratory syncytial virus (RSV), dengue, and hepatitis A, B, and C [23,24,25,26,27,28,29,30]. Among the different types of T cells, cytolytic CD8+ T cells can directly eliminate infected cells, whereas CD4+ helper cells activate B cells and recruit other innate immune cells via cytokine release. In a mouse model of influenza, populations of both these T cell types have been demonstrated to be essential for effective viral clearance and protection against lethal disease. While eliminating one T cell population results in only a slight delay in viral clearance, the elimination of both CD4+ and CD8+ cells significantly increases mortality [31]. In terms of immune memory, T cell responses are generally slower to develop but tend to be more cross-protective across different strains and variants. For example, among SARS-CoV-1 survivors, a significant T cell response to the virus has been observed to persist for up to six years after infection [32]. In contrast, although emergent SARS-CoV-2 variants show significant immune escape with respect to the B cell response, T cell responses remain relatively stable across different variants [33,34,35]. In the case of dengue, CD8+ T cells have been shown to provide cross-protection against multiple serotypes [36], while secondary influenza infections are associated with a notably faster expansion of T cells in the lungs and more rapid viral clearance [37]. Given the pivotal role of T cells in protection against viral infections, any dysregulation of these cells, especially in specific sub-populations, can have detrimental effects under certain circumstances. For example, a deficiency in Th17 cells has been shown to result in severe lung inflammation in a mouse model of influenza [38]. Similarly, in infections with the murine coronavirus strain JHM, regulatory T cells are essential for preventing demyelination in chronically infected mice [39]. There is substantial clinical, in vitro, and in vivo evidence to indicate the efficacy of T cell immunity in the context of viral clearance and protection against viral disease.

In contrast, the protection conferred by antibodies remains less clear. For example, with the exception of enteroviruses, it has been established that individuals with the X-linked agammaglobulinemia, who are unable to produce antibodies, do not show increased susceptibility to viral diseases [40]. Furthermore, in immune-naïve individuals infected with COVID-19, higher IgG concentrations correlate with a more severe disease course [26,41]. Moreover, treatment with convalescent serum containing the antigenic antibodies has been ineffective against both influenza and COVID-19 [41,42]. One plausible explanation for the poor efficacy of antibodies in cases of natural infections is the low levels of antigen-specific antibodies during early infection [43]. In many viral infections, including influenza, COVID-19, and dengue, the viremia often begins to decline between days 3 and 10 after the onset of symptoms, before a significant antibody response develops [44,45,46].

Antibodies can also have pathogenic effects. For example, in dengue, antibodies have been shown to enhance infection and disease (antibody-dependent enhancement) at sub-neutralizing titers via Fc-receptor interactions. Recent studies have demonstrated that afucosylated IgG1 antibodies, characterized by the absence of fucose residues on the Fc region, exhibit high affinity for the FcγIIIa (CD16a) receptor, which correlates with enhanced disease severity in mouse models [47,48]. Notably, nanobodies that block the binding of afucosylated IgG1 to CD16 can ameliorate disease progression [49]. Given that certain viral epitopes may be similar to those of the host organism, viruses can effectively evade immune responses via molecular mimicry. Accordingly, it is plausible that if antibodies are elicited against such epitopes, it might have detrimental autoantibody effects. However, molecular mimicry alone may not be sufficient to trigger autoimmune diseases [50].

### 2.3. Vaccine-Induced Immunity

The adaptive immune response is the primary mechanism through which vaccines confer their protective effects. Moreover, the relative importance of different T cell populations varies across viral infections. For instance, in a study of hepatovirus A infection in chimpanzees, the CD4+ response was associated with protection and significantly contributed to viral clearance [25]. In contrast, following T cell vaccination of mice against SARS-CoV-1, the CD8+ response alone was sufficient for protection [51]. Similarly, while T cells play important roles in viral clearance during RSV infection, skewing the T cell population toward a Th2 phenotype has been linked to severe inflammatory pathology [28]. This Th2 skewing has also been observed in SARS-CoV-1 vaccine trials, where it was associated with severe immunopathology [52]. Certain autoimmune reactions to COVID-19 vaccines, such as autoimmune hepatitis and myocarditis, have been correlated with the expansion of both CD4+ and CD8+ T cells [53,54]. Additionally, hyperactivation of T cells in response to COVID-19 infection may contribute to severe inflammation [55].

Despite the uncertainties surrounding the role of antibodies in natural infections, there is compelling evidence for their protective effects in the context of vaccination and antibody therapies, where much higher antigen-specific antibody concentrations can be achieved. For example, antibody titers correlate strongly with protection against both COVID-19 and influenza, reflecting vaccine efficacy [56,57]. Antibodies contribute to protection through neutralization, suppression of infection, and the activation of secondary immune effectors via the Fc domain. These mechanisms include antibody-dependent cellular phagocytosis and antibody-dependent cellular cytotoxicity (ADCC), which are associated with viral clearance [43]. Furthermore, antibodies can activate the complement system and neutrophils through antibody-dependent complement deposition.

The protective mechanisms of antibodies differ among viral strains. Mouse models of influenza vaccination and antibody therapy suggest that Fc-mediated functions, particularly ADCC, are critical for protection [58,59]. In contrast, complement system involvement varies; for instance, complement knockout mouse models of influenza exhibit higher mortality and morbidity, underscoring the complement system’s protective role in this context [60]. However, some experimental influenza vaccines, such as those targeting the M2 protein, act independently of the complement system [58]. In humans, complement deficiency increases the risk of bacterial infections but does not significantly affect susceptibility to viral diseases [61]. Similarly, studies of COVID-19 mRNA vaccination and antibody therapies have shown that Fc-mediated functions are critical for protection [62,63]. In contract, a study in mice revealed that antibody-mediated protection against COVID-19 depended on alveolar macrophages rather than neutrophils or the complement system [63].

In addition to adaptive immune responses, evidence indicates that live-attenuated vaccines (LAVs) can induce epigenetic changes in the innate immune system, leading to a trained immunity. These epigenetic modifications, such as histone modifications, DNA methylation, and non-coding RNA regulation, alter gene expression without changing DNA sequences. For example, specific histone marks like H3K4me3 (trimethylation of histone H3 at lysine 4) and H3K27ac (acetylation of histone H3 at lysine 27) have been shown to increase after pathogen exposure, enhancing accessibility to genes critical for inflammatory responses [64]. Such mechanisms are believed to underlie the non-specific health benefits observed with LAVs, including those for measles, Bacillus Calmette-Guérin (BCG), and polio [65]. These findings illustrate the varied roles of antibodies, T cells, and innate immune mechanisms in vaccine-induced protection, with differences observed across pathogens and vaccine platforms. This diversity highlights the complexity of immune responses to vaccination and underscores the need for further investigation to optimize efficacy and safety.

## 3. Vaccine Platforms for RNA Viruses

Humans are continually striving to develop effective strategies for the control of RNA viruses that are responsible for numerous diseases, including the influenza virus, dengue virus (DENV), human immunodeficiency virus (HIV), and SARS-CoV-2 [66]. The rapid rates at which RNA viruses mutate pose a perpetual threat to public health, necessitating the development of innovative and effective vaccine platforms. However, although traditional vaccines have provided a foundation for disease prevention, the ever-evolving nature of RNA viruses necessitates continual advances in modern vaccine technologies that will contribute to providing more rapid and adaptable responses to emerging viral threats, thereby ensuring better protection from and control of outbreaks. The development and implementation of these diverse vaccine strategies will be essential for maintaining global health and combatting future pandemics [67,68]. Whereas vaccine platforms constitute the core mechanisms underlying the generation of the immune response, it is the vaccine formulations, which encompass the combination of constituents, including the antigen, adjuvants, stabilizers, and delivery systems, that contribute to optimizing stability, delivery, and efficacy [69]. This is particularly important for the manufacture of RNA virus vaccines, as effective formulations can enhance immune responses, ensure stability against rapid mutation, and improve delivery, ultimately enhancing protection against emerging viral threats.

### 3.1. Traditional Approaches

Traditional vaccine platforms, such as inactivated and LAVs (Figure 1), have long played a central role in controlling RNA viral diseases. Inactivated vaccines are made from pathogens that have been killed or inactivated, such that they can no longer cause disease, whereas LAVs are made from weakened forms of a virus that although still capable of replication, do not cause severe illness. By mimicking natural infections, these approaches have proven effective against numerous RNA viruses.

#### 3.1.1. Inactivated Vaccines

The first inactivated vaccine used for the control of an RNA virus was the inactivated rabies vaccine developed by Louis Pasteur and Emile Roux in 1885. Pasteur inactivated the virus by drying the spinal cords of infected rabbits, which effectively neutralized the virus, whilst retaining its immunogenic properties [71]. Inactivated vaccines are developed by cultivating the target virus in culture, followed by inactivation of the virus using chemical agents, such as formaldehyde, or by applying heat. Formaldehyde inactivation is based on the cross-linking of viral proteins and nucleic acids, thereby effectively neutralizing the capacity of the virus to replicate, whilst preserving its structural integrity. Thus, although rendered non-infectious, the virus retains its antigenic structure, enabling it to stimulate an immune response [72]. Inactivated vaccines have been developed to target a diverse range of RNA viruses and can provide significant protection against numerous pathogens. For example, inactivated viral vaccines are used against influenza to induce cross-protective CD8+ cytotoxic T lymphocytes that confer immunity against multiple strains [73]. In the case of the rabies virus, inactivated vaccines have been found to prevent infection, with recent advances contributing to an enhancement of their immunogenicity, based on the expression of dendritic cell-targeting proteins via recombinant virus technology [74]. Similarly, inactivated vaccines are being assessed for the control of emerging RNA viruses, such as Zika, and have been shown to protect pregnant marmosets from Zika infection [75]. Similarly, Covaxin, a beta-propiolactone inactivated vaccine developed by Bharat Biotech, has shown promising results in clinical trials for the prevention of COVID-19 [76]. Additionally, advances in dengue vaccine development have shown that combining inactivated vaccines with specific immunogenic proteins (non-structural protein 3, NS3) enhances immune responses [77]. These examples serve to highlight the broad applicability of inactivated vaccines in controlling and preventing different RNA viral infections. These vaccines offer several advantages in terms of safety and stability. As the viruses used in these vaccines are inactivated, they are unable to replicate, and are thus generally considered to be safe for administration to immunocompromised individuals [78]. In addition, compared with LAVs, inactivated vaccines are more stable, as they do not require cold chain conditions, and are thus more convenient for distribution in regions with limited infrastructure [79]. Furthermore, these vaccines can elicit a strong immune response, as evidenced by the whole-virion inactivated SARS-CoV-2 vaccine, which induces a Th1-biased immune response and high levels of specific T lymphocyte response [80].

However, inactivated vaccines do have certain limitations. Notably, multiple doses are often necessary to achieve and maintain sufficient immunity. For example, studies have revealed that booster doses are necessary to sustain immunity, particularly against emerging variants, such as the SARS-CoV-2 Omicron variant [81,82]. Moreover, compared with other vaccine platforms, inactivated vaccines may induce a weaker immune response, necessitating adjuvants or supplementary doses. Additionally, these vaccines may not provide sufficiently broad cross-protection against different strains or variants of a given virus, as they primarily target specific viral antigenic components, thereby potentially limiting their efficacy against evolving pathogens [83,84]. Recent efforts that have sought to enhance inactivated vaccine platforms have focused on optimizing the manufacturing process. For example, cell culture systems have been improved by using suspension cultures and perfusion bioreactors that are efficient and conducive to the production of higher viral yields [85,86]. In addition, novel inactivation methods, such as the use of hydrogen peroxide, UV irradiation, and gamma irradiation, are being evaluated as alternatives to traditional chemical inactivation [87,88,89]. Moreover, with the aim of enhancing vaccine immunogenicity, recent research has focused on next-generation adjuvants, including TLR agonists and nanoparticle-based formulations, such as biocompatible calcium phosphate, gold, and silica particles [90,91]. However, although these innovations aim to enhance the safety and stability of inactivated vaccines, their adoption in licensed vaccines will require a thorough regulatory review and extensive testing.

#### 3.1.2. Live-Attenuated Vaccines (LAVs)

The first LAV against an RNA virus was the yellow fever vaccine developed by Max Theiler in 1930. Theiler attenuated this virus via serial passage from mouse embryo to chicken embryo tissue, with genetic mutations accumulating over the course of serial passage [92]. LAVs are developed via a multi-step process designed to generate weakened viruses that although unable cause disease, can still induce robust immune responses. The process is initiated by selecting and isolating pathogenic virus strains, followed by adaptation to cell cultures or other mammalian hosts, which contributes to the accumulation of mutations that reduce viral virulence [93]. Recent approaches have included genetic engineering techniques, including reverse genetics, designed to introduce targeted mutations and optimize codons, thereby directly attenuating the virus and enhancing vaccine safety and efficacy [94]. Having de-optimized the virus, it undergoes rigorous screening and validation via in vitro and animal model studies to ensure that it does not cause disease and retains its immunogenic properties [95]. In addition, ensuring the genetic stability of the attenuated virus is essential for preventing reversion to a more virulent form. LAVs have been evaluated for their efficacy against a range of RNA viruses, including the chikungunya virus, influenza virus, dengue virus, and SARS-CoV-2, offering several advantages, although also having some notable disadvantages [96,97,98,99]. One of the primary advantages of LAVs is their ability to induce a strong and enduring immune response based on closely mimicking natural infections. This results in broad and durable immunity, often requiring only a single or two doses [100]. Additionally, LAVs are known for their ability to exploit both humoral and cellular immune responses, thereby providing protection against multiple viral antigens [101]. However, LAVs also have certain limitations. Notably, from a safety perspective, there remains the possibility that attenuated viruses could revert to a more virulent form, particularly in immunocompromised patients [102,103]. Given that RNA viruses are mutation-prone, one study has shown that pre-existing immunity limits live-attenuated influenza vaccines’ ability to generate or boost T cell responses, particularly by hindering the formation of de novo T cell populations [104]. Moreover, LAVs require careful storage and handling, often necessitating storage at low temperatures, which limits their accessibility to resource-limited areas. Studies have, nevertheless, shown that apart from freeze drying or foam drying, using an appropriate stabilizer, such as trehalose-based stabilizers, can sufficiently increase stability [105]. Given these challenges, further investigations are required to identify suitable means of enhancing the safety and stability of LAVs, particularly to minimize the risks of reversion and improve their usability in diverse global settings.

### 3.2. Modern Approaches

By building on the foundations established by traditional vaccine platforms, modern vaccine approaches have contributed to significant advances in our ability to combat RNA viruses. Although traditional vaccines, including inactivated and LAVs, can provide essential protection against numerous RNA viral diseases, the rapid evolution and mutation of RNA viruses have driven the need for more innovative solutions [106,107]. Modern strategies, including subunit, viral vector, nucleic acid, and self-amplifying RNA vaccines, offer more adaptable and scalable options (Figure 2). Moreover, these approaches are designed to enhance the efficacy, safety, and speed of production, while also addressing the limitations of traditional methods. As RNA viruses continue to evolve and present new challenges, the development of novel vaccine designs will become increasingly important for maintaining global health and combatting future pandemics.

#### 3.2.1. Subunit Vaccines

Subunit vaccines use viral proteins or peptides to stimulate immune responses in the absence of living pathogens. These vaccines include virus-derived components, such as surface proteins or peptides, which are commonly recognized by the immune system and stimulate both humoral and cellular immunity. This approach facilitates targeted immune responses with a reduced risk of side effects associated with whole-virus vaccines. In this regard, surface proteins or peptides have been selected as ideal antigens and are generally membrane-integrated proteins in the case of enveloped viruses and capsid proteins for non-enveloped viruses, which are unstable when solubilized. The stabilization of proteins or peptides in subunit vaccines is important for maintaining their structural integrity and enhancing immunogenicity. One approach in this respect is molecular clamping, in which a synthetic peptide is fused to a viral protein to maintain the protein in its native conformation and prevent unfolding. By stabilizing the protein in this manner, the molecular clamp ensures that the antigen remains in a form that is most likely to be recognized by the immune system, thereby enhancing vaccine efficacy [108]. A further method involves protein engineering, such as the fusion of antigens with the Fc fragment of human IgG or the generation of mutations that contribute to structural stabilization [109]. For example, cross-linking agents, including 1-ethyl-3-(3-dimethylaminopropyl)-carbodiimide [110] and 3,3′-dithiobis(sulfosuccinimidyl propionate), have been used to generate covalent bonds between protein molecules, thereby maintaining their structure under different conditions [111,112]. In addition, by focusing on three-dimensional neutralizing epitopes that can be designed to avoid autoimmune responses, peptide-based vaccines can offer further alternatives. Moreover, peptides can be readily scaled up and modified in the event of identifying of a new mutant variant strain [113].

However, to ensure efficacy, subunit vaccines also require associated delivery systems for administration. Numerous delivery methods have been accordingly developed with a view to enhancing the stability, immunogenicity, and specificity of these vaccines. For example, lipid nanoparticles (LNPs) are commonly used to encapsulate and deliver proteins or peptides as antigens in subunit vaccines. These nanoparticles can protect against antigen degradation and enhance their delivery to antigen-presenting cells [114]. In addition, polymer-based nanoparticles, such as poly(lactic-co-glycolic) acid, have also been used, given their tunable properties and sustained-release capabilities [115]. In addition, virus-like particles (VLPs), which structurally mimic viruses but do not carry genetic material, thus rendering them non-infectious, have been used in subunit vaccines to present antigens in a manner that closely resembles that of natural viruses, thereby enhancing immune responses [116,117].

The recent integration of artificial intelligence (AI) in structural biology has contributed to a significant acceleration in the development of subunit vaccines. Among the approaches adopted to date, the use of AlphaFold AI tools has substantially enhanced the speed with which the three-dimensional structures of viral proteins and their biomolecular interactions can be predicted, thereby enabling researchers to identify potential epitope-specific portions of the virus that can be targeted by the immune system [118,119]. In turn, by ensuring that the selected viral proteins or peptides are optimally structured to elicit strong immune responses, these developments have facilitated the design of more effective vaccines. As AI continues to evolve, its role in vaccine development is expected to grow in prominence, contributing to a more rapid and accurate vaccine design processes [120,121].

#### 3.2.2. Viral Vector Vaccines

The use of viral vector vaccines dates back over 30 years, with the first successful application of a vaccinia virus vector for hepatitis B immunization, in which a hepatitis B surface antigen gene was inserted into simian virus 40 [122]. Since then, a range of viruses have been engineered to generate viral vaccine vectors, including retroviruses, lentiviruses, adenoviruses, adeno-associated viruses, cytomegaloviruses, and Sendai viruses [123,124,125,126,127]. These vaccines utilize viral vectors to deliver genetic material from a pathogen into host cells, which is then expressed to stimulate an immune response that mimics a natural infection [128]. The importance of viral vector vaccines was particularly highlighted during the COVID-19 pandemic, with the production of adenovirus-vectored vaccines, such as those developed by a collaborative venture between Oxford University and AstraZeneca plc. (ChAdOx1 nCoV-19), Johnson & Johnson (JNJ-78435735, Ad26.COV2.S), Gamaleya Research Institute of Epidemiology and Microbiology (Sputnik V), and CanSino Biologics lnc. (Convidecia), which have played pivotal roles in global vaccination efforts. These vaccines encode the spike protein of the SARS-CoV-2 virus that induces robust immune responses, and they have shown demonstrable efficacy against a number of COVID-19 variants [129].

One of the primary advantages of viral-vectored vaccines is their ability to generate strong immune responses at lower doses than those of traditional vaccines. Moreover, they are also relatively stable and do not require ultra-cold storage, making them accessible in regions with limited infrastructure [130]. As part of the continuing maturation of viral vector vaccine technology, the Coalition for Epidemic Preparedness Innovations has introduced a “100-day mission”, with the aim of developing and deploying effective vaccines within 100 days of identifying a new pathogen [131]. However, despite the value of these vaccines, they do have certain disadvantages, notably the potential for pre-existing immunity to the viral vector, which can contribute to reduced vaccine efficacy [132]. Additionally, over the course of the COVID-19 pandemic, billions of individuals were vaccinated, some of whom have developed mild-to-moderate side effects, thereby highlighting the need for the more extensive monitoring of long-term safety [133].

#### 3.2.3. Bacterial Vector Vaccines

A further potential delivery system that can serve as a means of ferrying DNA vaccines entails the use of live bacterial vectors, whereby genetically modified bacteria can be administered intranasally, orally, or intravaginally, thus offering multiple delivery routes. The innate immune response induced by the pathogen-associated molecular patterns of bacteria, as well as the potential adaptive immune responses they stimulate, mean that such modified bacteria serve not only as a vehicle conveying the DNA vaccine but also function as effective immunostimulatory adjuvants. To date, numerous bacterial strains have been extensively evaluated for their potential application as DNA vaccine vehicles, thereby providing a well-characterized genetic background and mutations for virulence attenuation [134,135,136,137]. Whereas the proposed DNA transfer mechanisms involve bacterial invasion, non-pathogenic commensal bacteria such as lactic acid bacteria (LAB) are generally considered more suitable for immunocompromised patients, although these may still pose a threat of potential invasive infections [138,139]. Given their resistance to acidic environments, LAB, including strains that are widely used as probiotics, can be delivered via the gastrointestinal route. Furthermore, attenuated pathogenic bacteria, such as strains in the genera *Salmonella*, *Listeria*, and *Mycobacterium*, have also shown promise as vectors for DNA transfer, in that they are capable of inducing both cellular and humoral immune responses, although there remains the possibility of reversion to the pathogenic state. Nevertheless, several live-attenuated bacterially vectored vaccines for viral infections have reached clinical trials, among which attenuated *Listeria monocytogenes* (BMB72) has been used to transfer the influenza A nucleoprotein [140,141], and *L. monocytogenes* (XFL-7) has been used to administer the rHPV-16 E7 antigen [142]. Additionally, Syrian hamsters have been immunized using *Salmonella*-mediated bactofection, a bacterially mediated form of genetic transfer, carrying replicon-based mRNA or attenuated *Francisella tularensis* platforms expressing structural proteins, which have been shown to provide protection against SARS-CoV-2 infection, with ameliorated lung pathology and efficacious antibody production [143,144]. These proof-of-concept findings could offer viable alternatives for needle-free vaccination, although further assessments are required.

#### 3.2.4. Nucleic Acid Vaccines

##### mRNA Vaccines

Nucleic acid vaccines, particularly mRNA vaccines, are groundbreaking vaccine platforms that are adaptable and can be readily modified in response to rapidly emerging viral epidemics/pandemics. mRNA vaccines deliver viral protein-encoded mRNA, such as that of the SARS-CoV-2 spike protein, into host cells. Therein, mRNA is subsequently translated to produce a viral protein that is displayed on the cell surface and triggers a viral protein-specific immune response. These vaccines have attracted considerable attention in the wake of the development of SARS-CoV-2 vaccines by Pfizer-BioNTech and Moderna, which were the first mRNA vaccines to be granted emergency use authorization from many countries during the pandemic. These vaccines have shown high efficacy, providing approximately 95% protection against COVID-19, as revealed in clinical studies, and have made a significant contribution in controlling the spread of SARS-CoV-2 [145]. Moreover, Drs. Katalin Karikó and Drew Weissman were honored with the Nobel Prize for their development of the mRNA vaccine [146,147,148,149]. Their work on this technology not only constituted a pivotal breakthrough in combatting COVID-19 but also represented a substantial advance in vaccine science.

Unlike traditional vaccines, which can take years to develop, one of the key advantages of mRNA vaccine technology is that these vaccines can be designed, manufactured, and produced within a comparatively short space of time, thereby making them ideal for responding to pandemics. Additionally, mRNA vaccines are flexible, as their sequences can be readily modified to target different variants of RNA viruses or even different viruses [150], which is an essential facet for combatting rapidly mutating RNA viruses such as SARS-CoV-2, influenza viruses, or mpox viruses, among which, there is a high likelihood of the emergence of novel variants that might evade pre-existing immunity [151,152,153].

Although mRNA vaccines have multiple advantages, their application does have certain drawbacks, of which a key disadvantage is that RNA molecules can lack stability and may be readily degraded. To overcome such problems, lipid nanoparticles are often used to encapsulate mRNAs, thereby protecting these molecules from degradation and facilitating delivery into host cells. A further disadvantage is their limited shelf life of less than 24 h, and thus the demand for stringent storage and transport conditions [154]. Nevertheless, researchers are striving to improve the stability of mRNA vaccines at higher temperatures, which will substantially contribute to reducing the costs and challenges of vaccine distribution [155]. New-generation platforms are currently being developed to address these challenges and to enhance the efficacy and accessibility of mRNA vaccines, among which are self-amplifying RNA vaccines, in which RNA molecules can be amplified within host cells, and circular RNA vaccines, which are more stable than linear mRNAs [156,157]. With these innovations, new-generation mRNA vaccines promise to reduce the dosage requirements and enhance the thermal stability of RNA vaccines, thus paving the way for the development of products that are more robust and easier to distribute globally [158]. Moreover, the potential application of mRNA vaccines extends beyond infectious diseases, for example, as therapeutics for cancers, autoimmune diseases, and other rare diseases [159,160,161]. Indeed, the triumphal development and deployment of mRNA vaccines during the COVID-19 pandemic have opened a window for further applications of this technology, in which it is predicted that AI will play a key role in optimizing and accelerating vaccine development, with the promise of a new era in vaccinology and personalized medicine [162].

##### DNA Vaccines

DNA vaccines, first conceptualized in the early 1990s, represent a novel approach to immunization using genetically engineered plasmid DNA to encode pathogen-derived antigens. Using this approach, engineered DNA is taken up by host cells, wherein the encoded antigen is synthesized and presented on the cell surface, thereby triggering both humoral and cellular immune responses [163,164]. Among the notable examples of this technology is ZyCoV-D, developed by Zydus, which was the first DNA vaccine to target COVID-19 and be approved for humans, and INO-4800, developed by Inovio, which has shown promise in clinical trials against SARS-CoV-2 as a safe booster [165]. The key advantages of DNA vaccines include their safety, as they do not involve live pathogens, and their stability, which often eliminates the need for refrigeration, making them easier to distribute in low-resource environments [166]. In addition, DNA vaccines can be rapidly developed in response to emerging RNA viruses. However, their use is not without limitations, among which is the relatively low immunogenicity in humans, which could be ascribed to the amounts used [167]. Different delivery methods, codon usage, and adjuvants are, nevertheless, being assessed in efforts to enhance the efficacy of these vaccines [168].

#### 3.2.5. Phage-like Particle Vaccines

Phage-like particle vaccines are an emerging technology that utilizes the structural properties of bacteriophages for displaying antigens from RNA viruses, thereby effectively stimulating a virus-specific immune response. A pioneering study of phage-like particles by George P. Smith in 1985 built a foundation for the broader application of this approach in vaccine development [169]. The Smith lab used a filamentous bacteriophage (M13) to display a foreign peptide in an immunologically accessible form. The concept of using bacteriophages for antigen delivery has been adopted in a number of formal vaccine development studies. For example, the lambda bacteriophage has been engineered to display the receptor-binding domain of SARS-CoV-2, which has been demonstrated to have high immunogenicity and elicit durable antibody responses. This phage-like particle vaccine not only offers protection against lung infections caused by coronaviruses but also highlights the potential application of phage-like particles as a versatile platform for vaccine development [170,171]. The advantages of phage-like particle vaccines include their capacity to elicit strong and targeted immune responses, owing to the multivalent display of antigens, as well as their stability and ease of production. However, a number of issues with their use are yet to be resolved, such as optimizing the expression of viral antigens on the phage surface and ensuring consistent immune responses among different populations [172]. Future research should focus on enhancing the design and delivery of these vaccines and examine the potential integration of synthetic biology and AI-driven optimization to enhance their efficacy and broaden their application in the treatment of infectious diseases.

#### 3.2.6. Novel Antigen Delivery Systems

In a constant effort to counter emerging and re-emerging infectious diseases, the development of novel antigen delivery systems is an ongoing pursuit that seeks to broaden the scope of protective immune responses. In this section, we describe studies that envision new strategies for vaccination with respect to LNPs and exosome-based carrier systems. LNPs are versatile platforms that enable the delivery of nucleic acid materials and liposomal drugs with potential clinical applications as antigen-presenting agents. Common liposome constituents include phospholipids, such as phosphatidylcholines, phosphatidylethanolamines, phosphatidylserines, and phosphatidylglycerols. These particles, which are generally 50–500 nm in diameter, mediate fusion with the cell membrane and the subsequent release of nucleic acid materials via endocytic uptake [173]. The use of LNPs in the distribution of SARS-CoV-2 mRNA vaccines developed by Pfizer-BioNTech and Moderna offered the promise of safe and efficacious delivery of this and other vaccines. In addition, LNPs have been approved for the delivery of small-molecule drugs for cancer treatment [174]. Liposomes can also be designed with surface-attached ligands to facilitate receptor binding or to present antigens directly [174,175]. In 2022, Sia et al. provided evidence for the use of LNPs as antigen-presenting agents that facilitated the presentation of recombinant influenza hemagglutinin on modified immunogenic liposome surfaces. These LNPs, containing cobalt porphyrin-phospholipid, can bind to His-tagged hemagglutinin trimers, and have been shown to confer protection against the influenza virus in animal models [176]. In their recent studies, these authors demonstrated that multivalent LNPs displaying both hemagglutinin and neuraminidase antigens on their surfaces elicit potent protection against several strains of influenza virus in different animal models, including mice and ferrets [177]. Exosomes, which are specialized lipid-bound extracellular vehicles derived from cells, ranging in size from 30 to 200 nm, represent a further promising approach to vaccine delivery [174,178] and can fuse with cells to facilitate the delivery of requisite cargoes, including peptides, lipids, proteins, and nucleic acids. The findings of several studies conducted to date have demonstrated the use of exosomes as potential vaccine delivery vehicles for the treatment of viral infections, including hepatitis B, HIV, and SARS-CoV-2 [179,180,181,182,183,184,185,186]. These vehicles also provide an alternative to LNPs, with fewer reported toxicity concerns [187]. Collectively, these strategies will pave the way for the development of the next generation of vaccines.

## 4. Case Studies of RNA Virus Vaccines

The development of vaccines against RNA viruses has advanced rapidly in recent years. Various vaccine platforms have proven to be effective in combating a wide range of RNA viruses, marking a new era in vaccine development. This section examines case studies of several RNA viruses, highlighting the different vaccine types developed, the challenges encountered during their development, and the technological innovations that have shaped the current landscape of RNA virus vaccines. A comparative overview of selected viruses is presented, highlighting their unique characteristics, current vaccine status, and the specific challenges encountered during vaccine development (Table 1).

### 4.1. SARS-CoV-2 (COVID-19): mRNA and Viral Vector Vaccines

The development of various types of vaccine was rapidly accelerated in response to the COVID-19 pandemic, among which are multi-antigen and internasal vaccines that are currently being developed with a view toward enhancing the immunogenicity/efficacy and ease of administration. A noteworthy example in this regard is a multi-antigen intranasal vaccine that provides board protection against a range of betacoronaviruses and alphacoronaviruses. This vaccine utilizes NanoSTING, a liposomally encapsulated endogenous stimulator of interferon gene agonist (STINGa, 2′,3′ cGAMP), combining target spike and nucleocapsid proteins, designed to promote a more comprehensive immunity and reduce the risk of immune escape [188]. In addition, a study has highlighted the importance of spike protein structural stability by employing the pentameric cholera toxin B subunit in conjunction with a self-assembling nanoparticle chassis, which contributes to influencing vaccine efficacy and provides a basis for the design of future inhaled vaccine platforms [189].

The most widely used vaccines against COVID-19 include mRNA vaccines such as the Pfizer-BioNTech (BNT162b2) and Moderna (mRNA-1273) vaccines. These vaccines, which were among the first to be developed and granted emergency use authorization during the COVID-19 pandemic, utilize LNPs to deliver mRNA encoding the spike protein of SARS-CoV-2 into host cells, wherein it is expressed and induces an immune response. The groundbreaking success of these vaccines was emphasized by the high efficacy and speed of development with which they were deployed during the pandemic, even in fields such as cancer [145]. Viral vector vaccines, such as AstraZeneca’s ChAdOx1 nCoV-19 and Johnson and Johnson’s JNJ-78435735, have similarly played pivotal roles in global vaccination programs. These vaccines utilize modified chimpanzee or human adenoviruses as vectors to deliver SARS-CoV-2 spike protein-encoded nucleic acids into host cells, and, notably, the stability of these viral vector vaccines ensures widespread accessibility in areas that lack −80 °C storage facilities [133].

Vaccine-induced side effects, particularly with adenoviral vector-based vaccines, have raised concerns. Severe thromboembolic events such as cerebral venous sinus thrombosis (CVST) and splanchnic vein thrombosis (SVT), often accompanied by thrombocytopenia, have been reported 4–14 days after vaccination. No such events have been observed with mRNA-based vaccines. This has led to the identification of vaccine-induced immune thrombotic thrombocytopenia, likely caused by an immune-based response to adenoviral vectors [190,191]. In contrast, mRNA vaccines, which have significantly reduced COVID-19 morbidity and mortality, have been associated with rare cardiovascular complications such as myocarditis, acute coronary syndrome, thrombosis, and hypertension. These events may be linked to angiotensin-converting enzyme 2 (ACE2) dysregulation caused by spike protein mimicry. Despite these rare risks, the benefit–risk ratio of mRNA vaccines remains highly favorable, with medical monitoring recommended for individuals with pre-existing cardiovascular conditions [192].

### 4.2. Influenza: Seasonal and Pandemic Influenza Vaccines

Influenza viruses, which are typically characterized by high rates of mutation, necessitate annual updates of vaccines to counter the prevailing circulating strains, with the World Health Organization [193] announcing recommendations for vaccine use. For the 2024–2025 season, a trivalent influenza vaccine that protects against three different influenza viruses (influenza A H1N1, H3N2, and B/Victoria lineage) was recommended to reduce the global burden of influenza viral infection-related illnesses [194]. To control the seasonal outbreaks of influenza, such as the H1N1 pandemic in 2009 and the H7N9 pandemic in 2013, it is essential to have measures in place that enable the rapid development and deployment of vaccines. In this regard, advances in the development of RNA vaccines, initially driven by the COVID-19 outbreak, are now being employed to develop novel or improved universal influenza vaccines that can be used against different variants of the influenza virus [195].

### 4.3. Dengue: Dengvaxia and Other Candidates

Dengue fever presents significant challenges for vaccine development, particularly with respect to the risk of antibody-dependent enhancement, which can occur when prior exposure to one of the four distinct serotypes increases the severity of infection with a different serotype [196]. In this regard, although the live-attenuated Dengvaxia (CYD-TDV) vaccine, approved in 2022, has shown efficacy in preventing dengue, it is only recommended for individuals with prior dengue virus infection. In individuals who have had no previous infection, vaccination could increase the likelihood of severe dengue fever or hospitalization if subsequently contracting a natural infection post-vaccination [197]. Moreover, in addition to necessitating serological testing prior to use, the vaccine needs to be administered in three doses and given 6 months apart [198]. Thus, although the Dengvaxia vaccine represents a significant breakthrough, its limitations highlight the need for alternative approaches. A major factor contributing to antibody-dependent enhancement is the emergence of fucosylated anti-dengue IgG1 antibodies [48,49,62], and studies have indicated that viral infections and LAVs can induce a higher rate of afucosylated antibody production [199]. In contrast, experience with COVID-19 vaccines has indicated that protein subunit and adenovector DNA vaccines do not produce afucosylated IgG1, whereas mRNA vaccines may only induce transient afucosylation [200,201]. These insights accordingly raise the possibility that employing protein or adenovector vaccination schemes for dengue, in combination with protein subunit vaccines, followed by mRNA vaccinations, could mitigate the generation of afucosylated anti-dengue antibodies. However, it remains uncertain as to whether vaccination against dengue antigens will promote responses comparable to those elicited by SARS-CoV-2 spike proteins during vaccination for COVID-19. Nevertheless, the more recent approval of Qdenga, a tetravalent chimeric live-attenuated vaccine, which, notably, does not appear to enhance antibody-dependent diseases [202], has further advanced the field. Collectively, these developments highlight the complexities and ongoing challenges in producing effective and safe dengue vaccines.

### 4.4. Zika: Current Status and Challenges

Zika virus, an emerging mosquito-borne virus that triggered a global health emergency in 2015–2016 due to its association with microcephaly (a developmental defect in the brains of infants), has spurred efforts to develop effective vaccines [203]. However, one of the difficulties in developing Zika vaccines is the requirement for long-lasting immunity, particularly in pregnant women, who have the highest risk for mother-to-infant viral transmission. Currently, multiple vaccine candidates are at different stages of development, including DNA, mRNA, viral vector, virus-like particle, inactivated virus, peptide-based, and recombinant protein vaccines [134]. However, a recent marked decline in Zika cases has significantly limited the occurrence of natural infections, thereby making it difficult to conduct clinical trials for the evaluation of vaccine efficacy. Nonetheless, some researchers are continuing to develop vaccines and initiate phase-1 clinal trials, such that a vaccine could be rapidly deployed in the event of a resurgence [204].

### 4.5. Enterovirus A71: EnVAX-A71

Enterovirus A71 (EV-A71) is a major cause of hand, foot, and mouth disease (HFMD), which is particularly prevalent in children below 5 years of age and can lead to severe neurological complications. In Asia, where the virus is most prevalent, the development of vaccines against EV-A71 has been a priority. The first inactivated EV-A71 vaccine using C4 genogroup strains was licensed in China and demonstrated high efficacy in preventing HFMD. EnVAX-A71 (an inactivated vaccine based on the B4 sub-genotype) provides an efficacy of at least 96.8% and is safe for administration to infants between 2 and 6 months of age [205]. Current efforts are focused on the development of next-generation vaccines such as EV-A71 VLP production using *Pichia pastoris*, which could provide broader protection, facilitate wider distribution, and potentially enable the incorporation of new adjuvants or delivery systems to enhance their efficacy [110].

### 4.6. Nipah Virus: Prospects and Opportunities

Given its capacity for human–human transmission and the associated high mortality rate reaching up to 75%, the Nipah virus is considered to have pandemic potential [206]. Currently, there are no approved vaccines or treatments for Nipah virus infections, and it has accordingly been identified as a high-priority pathogen according to the World Health Organization and Centers for Disease Control and Prevention [197,207,208]. Recent advances in Nipah virus vaccine research have identified several promising candidates, among which is a recombinant vesicular stomatitis virus (rVSV)-based vaccine that has been demonstrated to elicit long-lasting immunity in non-human primates, with 100% protection maintained even a year after vaccination [209]. Additionally, both the ChAdOx1 NiVB vaccine, developed using a viral vector vaccine platform, and mRNA-1215, an mRNA vaccine platform developed by Moderna, are currently undergoing clinical trials. The ChAdOx1 NiVB vaccine has been shown to provide full protection in hamsters and African green monkeys, with robust immune responses and no viral replication [210,211], whereas the mRNA Nipah vaccine has been found to elicit effective antibody responses in pigs, thereby prompting its ongoing evaluation for human use [212].

### 4.7. Comparison of Vaccine Platforms Against Selected RNA Viruses

To provide a clearer comparison of vaccine platforms, we have created a summarized table highlighting the different platforms used for developing vaccines against selected RNA viruses (Table 2). This table presents an overview of the traditional and modern vaccine approaches, including inactivated, live-attenuated, mRNA, DNA, and viral vector vaccines, and their applications in combating viruses such as SARS-CoV-2, influenza, Zika, and dengue. It serves as a concise reference to understand the strengths and limitations of each platform in the context of these viral threats.

## 5. Challenges and Opportunities in RNA Virus Vaccine Development

Challenges invariably bring opportunities, and the development of RNA virus vaccines is no exception. The rapid evolution of RNA viruses, driven by constant mutations, such as antigenic drift and shift, presents innumerable challenges in that it necessitates frequent vaccine updates in the continual drive to maintain efficacy [213,214]. Moreover, these challenges have also propelled the development of cutting-edge vaccine technologies, such as mRNA and self-amplifying mRNA vaccine platforms, which offer novel approaches for initiating swift responses to viral changes.

RNA viruses, such as influenza virus and SARS-CoV-2, are known for their rapid evolution, driven by mutations via mechanisms such as antigenic drift and shift [215,216,217]. Antigenic drift refers to small changes in viral surface proteins that accumulate over time, potentially reducing vaccine efficacy. At the time of writing, there have been more than 20 human SARS-CoV-2 variants since October 2020, with thousands of genome sequence submissions to the Global Initiative on Sharing All Influenza Data (GISAID), which includes coronavirus data obtained during the COVID-19 pandemic [218]. In contrast, antigenic shift involves more pronounced changes in sequence, often leading to new viral strains and pandemics, such as the H1N1 influenza outbreak in 2009. There are potentially 144 different sub-types of the influenza A virus associated with antigenic shifts among 18 different hemagglutinin types and 9 different neuraminidase types [219]. This is particularly evident in influenza and SARS-CoV-2, for which constant viral mutations necessitate frequent vaccine updates [220,221]. However, these challenges have fueled research on broad-spectrum and universal vaccines that can target a wider range of viral variants, thereby potentially reducing the need for constant modification [222,223]. A further important aspect in this context is the necessity to achieve a balance between immunogenicity and safety [224]. Effective vaccines induce strong immune responses without causing adverse events or harmful side effects, and in this regard, innovations such as protein subunit vaccines, VLPs, and viral vector vaccines are leading the way by offering safer, non-replicating options that still ensure a robust defense against evolving RNA viruses [225,226,227,228].

For numerous viruses, distribution and access are major barriers, particularly in low-resource settings. Many RNA vaccines, including mRNA-based vaccines, require ultracold storage, which poses logistical challenges in regions with limited infrastructure [229], as was prominently highlighted during the course of the global response to COVID-19, and has accordingly prompted rapid advances in vaccine stability and transport systems designed to overcome these difficulties [230]. Improving access to vaccines in underserved areas will be essential for enhancing the efficacy of global public health efforts [231]. Finally, a lack of public trust and vaccine hesitancy, which are often driven by misinformation and fear, have become significant barriers to achieving high vaccination rates, and addressing this issue will require transparent communication, robust public education, and community engagement to build confidence in the safety and efficacy of vaccines [232]. This challenge, exacerbated by the spread of misinformation on social media, highlights the need for better public health outreach strategies to dispel fears and hesitancy, and thereby ensure broad vaccine acceptance [233].

Nevertheless, despite these multiple challenges hindering vaccine development, scientific innovation has found and will continue to find effective solutions. Addressing these difficulties requires not only adaptation but also breakthroughs in vaccine technology. Recent technological and scientific advances, particularly in nanotechnology and cryo-electron microscopy, have significantly improved RNA viral vaccines. Nanotechnology has facilitated considerable progress in vaccine delivery by enabling the precise engineering of nanoparticles, such as LNPs, which are essential for stabilizing mRNA vaccines and facilitating cellular uptake, as demonstrated in COVID-19 vaccines [234,235]. Moreover, nanoparticles can be designed to mimic viral structures, thereby enhancing immune system recognition and response [235]. In addition, by resolving viral protein structures at near-atomic resolution, enabling the detailed mapping of essential antigens, such as spike proteins and cryo-electron microscopy, has further advanced vaccine design [236]. This technology not only clarifies complex protein structures but also aids in improving pre-fusion-stabilized proteins, thus optimizing these for immune recognition and enhancing vaccine efficacy against rapidly evolving pathogens [237]. Other developments include ongoing advances in adjuvants and delivery systems that will contribute to boosting vaccine efficacy by enhancing the immune response at smaller doses, thereby facilitating more efficient protection [238]. An example in this field is graphene oxide (GO), a water-soluble single-layer carbon that spontaneously adsorbs different antigens. As an adjuvant, GO recruits antigen-presenting cells and induces both cellular and humoral immune responses [239,240]. Among the next generation of adjuvants, researchers have developed small-molecule immune potentiators as TLR7 agonists and optimized these using aluminum salts (alum) for vaccines. For example, the TLR7-based adjuvant AS37 has been shown to elicit enhanced immunogenicity in animal studies and has now entered clinical evaluation, with results to date indicating a good safety profile and effective immune activation in humans, consistent with earlier findings in non-human primates [241,242]. As novel delivery methods, nasal vaccines against COVID-19 are now explored in phase 1 trials sponsored by the National Institutes of Health, aiming for higher efficacy and reducing the need for needles, which will predictably contribute to improving compliance with existing vaccines [243,244]. These innovations not only address current issues but also bolster our preparedness for future pandemics.

## 6. Future Directions and Emerging Strategies

The future of RNA virus vaccine development is moving toward more comprehensive and adaptable solutions, with the aim of providing broad protection against different viruses or strains of viruses. Among recent developments, Moderna has announced the completion of phase 3 clinical trials for a single vaccine targeting both SARS-CoV-2 and influenzas viruses. Among the different strategies being assessed, the mRNA vaccine platform facilitates rapid development and flexibility in response to changes in the target virus. In addition, Moderna is attempting to incorporate antigens of a third virus, the RSV, into the dual COVID-flu vaccines, and maximize the number of antigen instructions that can be encapsulated within an LNP [245]. Moreover, by targeting the conserved regions that remain stable among different mutants, a universal vaccine could provide long-term immunity and reduce the frequency of vaccination. In this regard, the potential application of a nucleoside-modified mRNA-lipid nanoparticle vaccine encoding antigens from 20 known influenza A subtypes and B lineages has been demonstrated. This multivalent vaccine was found to induce strong, cross-reactive, and subtype-specific antibodies in both mice and ferrets, thereby providing protection against a wide range of viral strains [245]. These findings accordingly provide compelling evidence to indicate the efficacy of mRNA vaccines in providing broad protection against antigenically diverse viruses, thereby offering a promising basis for future pandemic preparedness.

The integration of AI into RNA vaccine development represents a transformative advance in the fields of immunology and infectious disease preparedness [246,247]. AI has the potential to enhance multiple aspects of vaccine design, including the development of personalized vaccines, optimization of vaccine formulations, and prediction of future pandemic viral strains. One application of AI in the development of RNA vaccines is in the production of personalized vaccines. By analyzing an individual’s genomic data, AI algorithms can identify specific genetic variants that may influence immune responses [248,249]. In addition, the MIT group has developed a machine learning-based design system, OptiVax, and a validation tool, EvalVax, which can be used to optimize and evaluate peptide vaccine formulations for SARS-CoV-2 [250,251]. This will contribute to the optimization of personalized vaccines to better complement an individual’s immune profile, thereby enhancing vaccine efficacy. By addressing factors such original antigenic sin, in which initial immune responses limit adaptability to new variants, and potential autoantigen triggers, personalized vaccines will offer more precise and effective protection. Furthermore, AI tools, such as machine learning and deep learning, will facilitate the design of vaccines against RNA viruses based on antigenic epitope prediction, vaccine candidate identification, and protein folding prediction [252,253,254,255]. Among the AI-driven platforms being assessed, Rapidly Adaptive Viral Response utilizes machine learning algorithms to assess different factors, including antigen selection and peptide presentation, thereby ensuring that the developed vaccines induce an effective stimulation of the immune system [256,257]. The combination of these innovations will improve the efficacy and safety of vaccines, particularly in the context of rapidly evolving pathogens and personalized medicine strategies.

In addition, the One Health approach highlights the importance of addressing the network of human, animal, and environmental health, recognizing that many RNA viruses have crossed the species barrier [258]. Health vaccinology is an emerging field that incorporates the principles of the One Health approach into vaccine development to address the interconnected health of humans, animals, and the environment. Health vaccinology focuses on multispecies vaccination strategies to control zoonotic diseases at their source, thereby preventing the transmission of pathogens from animals to humans [259,260]. By utilizing advanced tools, such as systems biology and computational models, researchers are enhancing vaccine design and improving the efficacy of vaccines in preventing zoonotic diseases. Notably, this strategy places particular emphasis on the importance of multisectoral collaboration, encompassing veterinary science, human medicine, and environmental monitoring, in addressing the complex dynamics of zoonotic diseases [193].

The development of RNA virus vaccines is advancing through the generation of multivalent mRNA vaccines, AI-driven designs, and the One Health vaccination approach. These innovations have facilitated the development of broad protection against diverse RNA viruses, personalized vaccines, and enhanced zoonotic disease control. Collectively, these advances represent a transformative step toward more effective, adaptable, and durable solutions for pandemic preparedness and future viral threats.

## 7. Conclusions

In recent years, significant progress has been made in the development of RNA virus vaccines, driven in particular by the challenges posed by the global COVID-19 pandemic. Different vaccine platforms, coupled with research on immune responses elicited by a viral infection or vaccination, have enabled us to gain a better assessment of vaccine efficacy and safety. Biomedical laboratories and pharmaceutical companies are focusing on developing single vaccines with the capacity to protect against multiple viruses or different strains of the same virus. Furthermore, the integration of AI is revolutionizing vaccine design and offering the prospect of personalized vaccine development and formulation adjustments based on different external factors.

Using demographic modeling and data from the Vaccine Impact Model Consortium (VIMC) for 110 countries, vaccinations administered between 2021 and 2030 are expected to avert 51.5 million deaths [261]. The integration of the One Health approach, which emphasizes the interconnectedness of human, animal, and environmental health, is essential for addressing zoonotic diseases, many of which are caused by RNA viruses. The development of RNA virus vaccines will continue to evolve in the future. With the potential for universal vaccines targeting conserved viral regions and AI tools capable of predicting future viral threats, the future holds considerable promise in providing more comprehensive and long-lasting protection. These advances will transform the global response to emerging and re-emerging RNA viruses, ensuring more rapid, targeted, and effective vaccines for combatting evolving threats.

## Figures and Tables

**Figure 1 vaccines-12-01345-f001:**
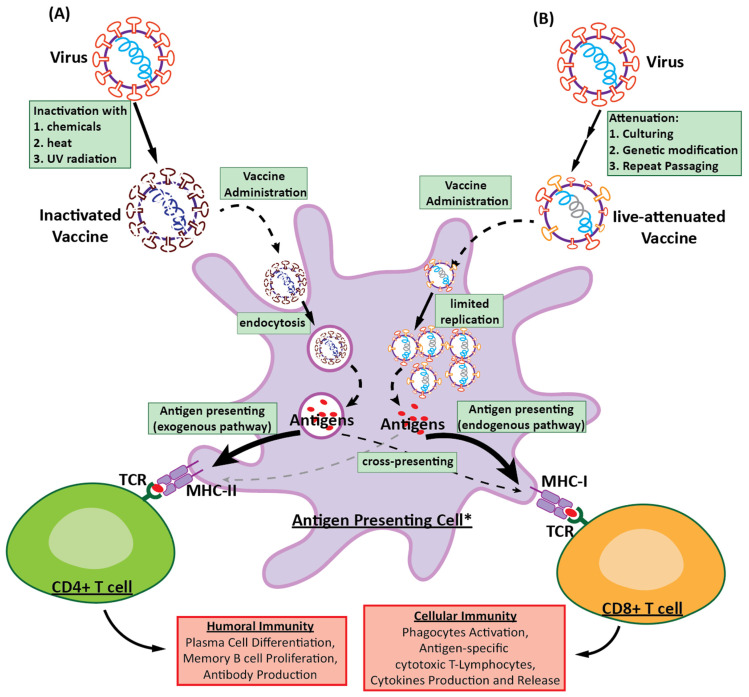
Schematic representation of traditional vaccines. (**A**) Inactivated vaccines are prepared by inactivating viruses with chemicals, heat-treatment, or UV radiation. (**B**) Live-attenuated vaccines are prepared using a weakened form of the virus, which is attenuated through methods such as non-human cell culturing, genetic modification, or repeated passaging. In contrast, inactivated vaccines contain killed or inactivated forms of the virus. After administration, inactivated vaccine antigens are taken up by antigen-presenting cells (APCs) via endocytosis. Inside APCs, these exogenous antigens are processed and displayed on MHC Class II molecules. The antigen–MHC-II complexes are presented on the surface of APCs, where they are recognized by T cell receptors (TCRs) on CD4+ T helper cells. This recognition triggers the activation of CD4+ T cells, which play a critical role in humoral immunity by assisting B cells in antibody production. Exogenous antigens can also be cross-presented to MHC-I. In comparison, live-attenuated vaccines infect host cells, including APCs and other host cells, but with limited replication, mimicking a natural infection. After infection, the antigens derived from the pathogens are processed internally by APCs, broken into smaller fragments, and presented on MHC Class I molecules. These MHC-I-antigen complexes are recognized by CD8+ cytotoxic T cells. This interaction activates CD8+ T cells, leading to their proliferation and enabling them to target and destroy infected host cells directly. Endogenous antigens can also be degraded and accessed to MHC-II molecules via autophagy and non-autophagic pathways [70]. * Note: Live attenuated vaccines are capable of infecting other host cells, mimicking natural infection processes.

**Figure 2 vaccines-12-01345-f002:**
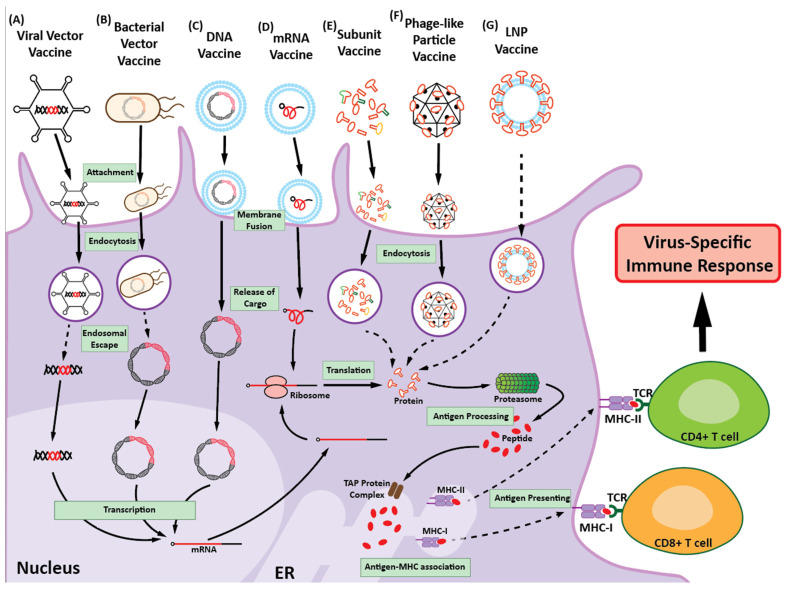
Schematic representation of modern vaccines. (**A**) Viral vector vaccines, which carry antigen-encoding DNA, attach to the host cell, undergo endocytosis, and release genetic material upon escaping endosomes. (**B**) Similarly, bacterial vector vaccines release their DNA cargoes into the host cell following attachment and endocytosis. (**C**) DNA vaccines encapsulated in lipid nanoparticles (LNPs) enter the cell and release the DNA into the cytoplasm. The DNA from viral vector, bacterial vector, or LNP-DNA vaccines subsequently enters the nucleus, wherein it is transcribed to mRNA. (**D**) mRNA vaccines directly deliver mRNA into the cytoplasm via membrane fusion, wherein ribosomes translate this to yield antigenic proteins. (**E**) Subunit vaccines and (**F**) phage-like particle vaccines deliver pre-formed antigenic proteins to host cells. (**G**) LNP vaccines containing antigenic proteins are taken up by cells. In all platforms, the produced antigenic proteins are processed by the endoplasmic reticulum (ER) and proteasomes to generate peptides, which bind to MHC-I or MHC-II molecules. These MHC–peptide complexes are presented on the cell surface, activating CD8+ and CD4+ T cells, thus leading to a virus-specific immune response.

**Table 1 vaccines-12-01345-t001:** Summary table for selected RNA virus.

Virus	Peculiarities	Vaccine Status	Challenges
SARS-CoV-2	Causes COVID-19 with symptoms ranging from mild to severe High transmissibility RNA virus with rapid mutation (notably in spike protein)	Several vaccines developed: mRNA (Pfizer, Moderna), viral vector (AstraZeneca, J&J), inactivated (Sinovac)	Variants reducing vaccine efficacy Need for booster shots
Influenza	Seasonal epidemics; segmented RNA genome enabling antigenic drift and shift Broad host range	Annual vaccines available: inactivated, live-attenuated, recombinant types	Mismatch in strain prediction Antigenic variation reduces long-term efficacy
Dengue	Four distinct serotypes (DENV-1 to DENV-4); reinfection with a different serotype can cause severe disease Spread by *Aedes* mosquitoes	Live-attenuated (Dengvaxia) approved for seropositive individuals in some regions, Tetravalent chimeric LAV (Qdenga)	Risk of antibody-dependent enhancement (ADE) in seronegative individuals
Zika	Linked to congenital Zika syndrome (microcephaly in infants) Transmitted by *Aedes* mosquitoes and sexually	No licensed vaccines available	Balancing immune response to avoid potential ADE (shared risk with Dengue) Difficult to conduct clinical trails
Enterovirus (EV-A71)	Includes polioviruses and EV-A71, causing polio, hand-foot-and-mouth disease, and encephalitis	Oral (OPV) and inactivated (IPV) vaccines effective for polio EV-A71 vaccines available in limited regions	Lack of globally available EVA71 vaccines Regional disparities in access to existing vaccines
Nipah	Zoonotic virus with high fatality rate Spread by fruit bats and through human-to-human transmission Neurological symptoms common	No licensed vaccines available (both viral vector vaccine and mRNA vaccine are undergoing clinical trials)	Sporadic outbreaks hinder largescale trials High-risk, small population targets complicate development

This table provides an overview of selected RNA viruses, highlighting their distinctive features and challenges in vaccine development. For each virus, the table includes a brief description of the vaccines currently available on the market, as well as specific obstacles encountered during vaccine development.

**Table 2 vaccines-12-01345-t002:** Summarizes the main features, similarities, and differences in the approaches to vaccination for different classes of RNA viruses.

Target Virus	Vaccine Platform	Key Features/ Approaches	Challenges	Similarities	Differences
SARS-CoV-2 (COVID-19)	mRNA (Pfizer Inc. New York, NY, USA/BioNTech SE. Mainz, Germany, Moderna Inc. Cambridge, MA, USA)	mRNA vaccines use LNPs to deliver spike protein encoding.	Rapid development and high efficacy. Need for ultra-cold storage for some vaccines	Induce strong immune responses.	Genetic material (mRNA, DNA, protein directly). Delivery mechanism (LNP, viral-vector, LNP). Antigen design. Rare side effects (myocarditis, VITT).
	Viral Vector (AstraZeneca plc Cambridge, UK, J&J Inc. New Brunswick, NJ, USA)	Viral vector vaccines use adenoviruses as vectors.			
	Multi-antigen LNP Vaccine	Liposome (stimulator of INF gene, spike protein, nucleocapsid protein), intranasal.	Manufacturing and scale-up, coldchain requirements.		
Influenza	mRNA	Annual updates to match circulating strains.	High mutation rates requiring frequent updates; development of broad-spectrum vaccines, cold chain requirements.	Both require frequent updates due to mutation rates, scalability.	Antigen (strain-specific antigen, targeting three strains, broadly conserved antigens). Protection (strain-specific, strain2-specific, broad protection). Antigen coverage.
	Recombinant subunits Vaccine	Seasonal (trivalent).			
	RNA-based Universal Vaccines	Development of universal vaccines targeting multiple influenza strains.			
Dengue	Live-attenuated (Dengvaxia by Sanofi, Paris, France)	Mimics natural infection.	Rick of serotype imbalance, antibody-dependent enhancement.	Risk of antibody-dependent enhancement (ADE); multiple serotypes complicate vaccine design.	Requires prior infection for safe vaccination
	Protein Subunit	Protein delivered with adjuvants.	Primarily humoral immunity, durability.		Primarily humoral immune response
	mRNA	LNP deliver mRNA encoding dengue antigens.	mRNA design against different serotypes.		Safer, fast development
	Adenovector	Adenovirus carrying dengue antigen infects cells.	Pre-existing immunity.		Balanced humoral and cellular immunity
	Chimeric Live-attenuated (Qdenga by Takeda Pharmaceuticals, Osaka, Japan)	Live-attenuated virus using a DENV-2 backbone.	Risk of replication, production complexity.		Strong humoral and cellular immunity, DENV-2 backbone
Zika	mRNA	mRNA encoding Zika virus antigens.	Ultra-cold storage limiting distribution, high cost.	Most platforms focus on envelop protein or prM protein, avoidance of components that could lead to antibody-dependent enhancement (ADE), vaccines are designed for pregnant women who are high-risk of Zika infection.	Focus on pregnant women and long-lasting immunity; Zika cases are rare, limiting trial opportunities.
	DNA	DNA plasmids encoding Zika antigens.	Limit immunogenicity.		
	Viral Vector	Viral vector to deliver Zika antigen.	Pre-existing immunity to the vector.		
	Virus-like Particle	Self-assembled viral proteins mimicking Zika infection.	Production complexity, adjuvant requirement.		
	Inactivated Virus	Non-replicative Zika virus.	Requires multiple doses/boosters.		
	Peptide-based	Short synthetic peptides representing Zika epitopes.	Limited T cell activation.		
	Recombinant Protein	Envelope or prM for use as antigen.	Requires potent adjuvants/high production cost.		
Enterovirus A71 (EV-A71)	Inactivated vaccine (EnVAX-A71 by Sinovac Biotech, Beijing, China)	Uses chemically or heat-inactivated whole enterovirus particles.	Requires large-scale virus cultivation, limited cellular immunity.	Non-replicative design, both require adjuvant.	Focus on infant and young children vaccination; different virus strain considerations (C4 vs. B4 genogroups).
	VLP Production	Composed of self-assembled structural proteins mimicking the virus without genetic material and adjuvant improvements.	May lack certain non-structural viral proteins, limiting T-cell responses.		
Nipah Virus	Recombinant Vesicular Stomatitis Virus (rVSV)	rVSV is engineered to express Nipah virus antigens (e.g., glycoprotein G) on its surface.	Potential side effects due to replication in certain hosts, slow development.	Nipah virus-targeting antigens, glycoprotein (G), or fusion protein (F), non-replicative in humans, scalable, single-antigen design which reduces risk of ADE.	Dosing requirements, safety considerations, speed of development, viral vector might have pre-existing immunity.
	mRNA (Moderna)	Delivers mRNA encoding Nipah virus glycoproteins.	No approved vaccines; high mortality rate, requiring urgent vaccine development.		
	Viral Vector (ChAdOx1 NiVB by University of Oxford, Oxford, UK.)	ChAdOx1 vector encoding Nipah virus antigens.	Pre-existing immunity.		

Summarization is based on the case studies presented in Section 4. This comparison will provide insights into the strategies employed for each virus, the technologies involved, and the key challenges associated with developing effective and safe vaccines.

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
