# Peer review of "Vaccine Strategies Against RNA Viruses: Current Advances and Future Directions"

_vaccines, 2024, doi:10.3390/vaccines12121345_

Round 1
Reviewer 1 Report
Comments and Suggestions for Authors
Manuscript "vaccines-3259432" by Kuei-Ching Hsiung et al. describes mechanism of innate and adaptive immunity, as well as traditional and advanced platforms for antiviral vaccine development. The manuscript also discusses the current status of antiviral vaccines and development pipelines, the application of new platforms to streamline the development process and the necessity for frequent vaccine updates against various RNA viruses and their variants. Lastly, the manuscript explores the integration of AI tools into vaccine design such as epitope prediction personalized vaccine.
The manuscript is an excellent review, providing a substantial amount of information for the field. It includes contents describing not only applications and pros and cons of both inactivated and live, attenuated vaccines developed traditionally, but also those of advanced vaccines such as vector vaccine, DNA vaccine, subunit vaccine, phage-like particle vaccine, mRNA vaccine and LNP as feasible and promising delivery system. Another interesting aspect is the case studies of antiviral vaccines against various RNA viruses such as SARS-CoV-2, Influenza, Dengue, Zika and et al., which update the direction and progress of vaccine development in these viruses.
I only have minor comments:
In lines 567 and 274, what is the disadvantages of inducing aflucosylated antibodies in the development of vaccine against Dengue. And why sequentially application of protein/vector subunit vaccine, and mRNA vaccine would mitigate the generation of aflucosylated antibodies and why this strategy is optimal compared to single strategy?
Author Response
Dear Reviewer,
Thank you for your thoughtful and detailed feedback on our manuscript. We appreciate your positive remarks regarding the comprehensive coverage of antiviral vaccine development and the various platforms discussed.
Comment 1:
In lines 567 and 274, what are the disadvantages of inducing afucosylated antibodies in the development of vaccine against Dengue. And why sequentially application of protein/vector subunit vaccine, and mRNA vaccine would mitigate the generation of afucosylated antibodies and why this strategy is optimal compared to single strategy?
Response 1:
Afucosylated IgG1 antibodies have been associated with an elevated risk of severe secondary dengue disease (Bournazos et al., 2021). In vivo humanized models indicate that afucosylated anti-Dengue IgG1 can lead to antibody-dependent enhancement (ADE) with increased mortality, a phenomenon not observed with fucosylated anti-Dengue IgG1 (Yamin et al., 2023). Furthermore, inhibition of afucosylated IgG1 interaction at its glycosylation site through a specific nanobody binding reduced the disease-enhancing effects, supporting the role of afucosylated IgG1 as a contributor to ADE (Gupta et al., 2023). Thus, current findings suggest that afucosylated anti-Dengue IgG1 constitutes a primary risk factor for ADE in secondary dengue infection.
Protein subunit and adenovector vaccines have been shown to induce nearly 100% fucosylated antibodies, as documented in studies on COVID-19 vaccines (Coillie et al., 2023b; Buhre et al., 2023; Reinig et al., 2024). This trend extends to protein subunit vaccines in hepatitis B and malaria, which have similarly been associated with high antibody fucosylation (Larsen et al., 2021a; Larsen et al., 2021b). Therefore, a vaccination strategy leveraging protein subunit and adenovector platforms may help mitigate ADE risk in dengue.
In contrast, mRNA vaccines against COVID-19 have been observed to induce transient afucosylation in immunologically naïve individuals, though not in those with prior antigen exposure (Coillie et al., 2023a; Buhre et al., 2023; Reinig et al., 2024; Kiszel et al., 2023). Should a similar response occur with dengue mRNA vaccines, an initial immunization with a non-mRNA vaccine could help prevent early afucosylation. Moreover, mRNA vaccine-induced IgG4 production—characterized by the lowest Fc-receptor affinity—presents a potentially favorable profile for dengue, where ADE relies on Fc-receptor-mediated entry (Irrgang et al., 2022; Buhre et al., 2023; Reinig et al., 2024).
Thank you for your thoughtful feedback. I appreciate your insights regarding the details mentioned. However, I believe that including this information within the main text would make it too detailed and could disrupt the flow of the manuscript.
I hope my response here clarifies your concerns, and I look forward to your further thoughts.
All the best,
Shin-Ru Shih
Research Center for Emerging Viral Infections, Chang Gung University
Department of Medical Biotechnology and Laboratory Science, College of Medicine, Chang Gung University
259 Wen-Hwa 1st Road, Kwei-Shan, Taoyuan 333, Taiwan
Tel: +886-3-2118800 ext. 5497
Fax: +886-3-2118174
Email: srshih@mail.cgu.edu.tw
Reference:
- Bournazos S, Vo HTM, Duong V, et al. Antibody fucosylation predicts disease severity in secondary dengue infection. Science. 2021;372(6546):1102-1105. doi:10.1126/science.abc7303
- Buhre JS, Pongracz T, Künsting I, et al. mRNA vaccines against SARS- CoV-2 induce comparably low long-term IgG Fc galactosylation and sialylation levels but increasing long-term IgG4 responses compared to an adenovirus-based vaccine. 2023;(January):1-18. doi:10.3389/fimmu.2022.1020844
- Coillie J Van, Pongracz T, Rahmöller J, Chen H, Geyer E, Vught LA Van. The BNT162b2 mRNA SARS-CoV-2 vaccine induces transient afucosylated IgG1 in naive but not in antigen-experienced vaccinees. EBioMedicine. 2023;87(December 2022):1-19. doi:10.1016/j.ebiom.2022.104408
- Coillie J Van, Pongracz T, Slim MA, et al. Comparative analysis of spike-specific IgG Fc glycoprofiles elicited by adenoviral, mRNA, and protein-based SARS-CoV-2 vaccines. iScience. 2023;26(107619). doi:10.1016/j.isci.2023.107619
- Gupta A, Kao KS, Yamin R, et al. Mechanism of glycoform specificity and in vivo protection by an anti-afucosylated IgG nanobody. Nat Commun. 2023;14(1). doi:10.1038/s41467-023-38453-1
- Kiszel P, Sík P, Miklós J, et al. Class switch towards spike protein specific IgG4 antibodies after SARS CoV 2 mRNA vaccination depends on prior infection history. Sci Rep. 2023;(0123456789):1-12. doi:10.1038/s41598-023-40103-x
- Larsen MD, de Graaf EL, Sonneveld ME, et al. Afucosylated IgG characterizes enveloped viral responses and correlates with COVID-19 severity. Science (80- ). 2021;371(6532). doi:10.1126/science.abc8378
- Larsen MD, Lopez-Perez M, Dickson EK, Ampomah P, Tuikue Ndam N, Nouta J, Koeleman CAM, Ederveen ALH, Mordmüller B, Salanti A, Nielsen MA, Massougbodji A, van der Schoot CE, Ofori MF, Wuhrer M, Hviid L, Vidarsson G. Afucosylated Plasmodium falciparum-specific IgG is induced by infection but not by subunit vaccination. Nat Commun. 2021 Oct 5;12(1):5838. doi: 10.1038/s41467-021-26118-w. PMID: 34611164; PMCID: PMC8492741.
- Irrgang P, Gerling J, Kocher K, et al. Class switch towards non-inflammatory IgG isotypes after repeated SARS-CoV-2 mRNA vaccination. Sci Immunol. 2022;4(December):2022.07.05.22277189.
- Reinig S, Kuo C, Wu C-C, Huang S-Y, Yu J-S, Shih S-R. Specific long-term changes in anti-SARS-CoV-2 IgG modifications and antibody functions in mRNA, adenovector, and protein subunit vaccines. J Med. 2024;(January). doi:10.1002/jmv.29793
- Yamin R, Kao KS, MacDonald MR, et al. Human FcγRIIIa activation on splenic macrophages drives dengue pathogenesis in mice. Nat Microbiol. 2023;8(8):1468-1479. doi:10.1038/s41564-023-01421-y
Reviewer 2 Report
Comments and Suggestions for Authors
In this review, the authors have attempted to summarize the available information on the development of vaccines against RNA-containing viruses, from the perspective of the past and the future. Despite the fact that a large body of literature was analyzed and summarized, the main idea of the review remained unclear, and the material was not systematized. It is not clear why RNA viruses of different families, which have completely different biology and different methods of vaccine preparation, should be lumped together in a single pile of RNA viruses. On the other hand, a lot of RNA viruses were left out of the picture. Also, the order in which the material is presented raises questions. For example, the authors present a set of very different facts about the B- and T-cell immune response to different viruses without systematizing the information in any way. The authors describe obvious facts about different vaccine platforms, while randomly selecting examples for some viruses, constantly jumping from one virus family to another. This makes it impossible to grasp the main idea of the review and to understand why all these disparate facts were presented. In addition, there are many grammatical errors and unrelated sentences in the article.
Comments on the Quality of English Languagethere are many grammatical errors and unrelated sentences in this review
Author Response
Dear reviewer,
The authors appreciate the detailed and thoughtful feedback, which provide valuable insights. We carefully considered the comment and believe the revised structure and scope of the review effectively deliver the intended purpose. The primary aim of this manuscript is to provide an overarching view of RNA virus vaccine development, highlighting both the common challenges and unique considerations across different RNA virus families, such as genetic variability and high mutation rates. This approach, encompassing a broad range of RNA virus families, focuses on cross-platform strategies that could benefit multiple virus types. We believe this perspective is essential in advancing vaccine research.
To specifically discuss the applications of different vaccine platforms, we focused on criteria such as epidemiological significance and vaccine development status, rather than aiming for exhaustive coverage. While we recognize the value of including more examples, we aimed for a balance between depth and scope to maintain readability and focus. To enhance this balance, we have added a section on Nipah virus, which further illustrates the applicability of vaccine strategies to high-priority RNA viruses without overwhelming the reader with extensive details on each virus.
Additionally, the order and structure of the material are intended to highlight emerging technologies, such as the integration of AI tools for epitope prediction and personalized vaccine design, to streamline the development process. The order and structure of the material are crafted to enable readers to see connections between different vaccine platforms and immune responses, thus justifying transitions between virus families. We sincerely appreciate the thoughtful suggestion to present these sections in a more segmented format. After careful consideration, we feel that the current flow helps maintain a cohesive narrative that best supports the paper’s objectives. We hope this approach aligns with the reviewer’s perspective.
We are grateful for the reviewer’s attention to detail regarding language and content. The manuscript has been sent to a professional English editing service to ensure it meets the journal’s standards. We believe these steps have addressed the concerns raised, and we sincerely appreciate the reviewer’s feedback, which we will continue to keep in mind.
In summary, we respectfully maintain the current structure and scope as they align with our intended goals for this review. We hope the reviewer can appreciate our perspective, and we trust that our rationale addresses the comments provided.
All the best,
Shin-Ru Shih
Research Center for Emerging Viral Infections, Chang Gung University
Department of Medical Biotechnology and Laboratory Science, College of Medicine, Chang Gung University
259 Wen-Hwa 1st Road, Kwei-Shan, Taoyuan 333, Taiwan
Tel: +886-3-2118800 ext. 5497
Fax: +886-3-2118174
Email: srshih@mail.cgu.edu.tw
Reviewer 3 Report
Comments and Suggestions for Authors
The review about vaccine advances against RNA viruses, including SARS-CoV-2 by Hsiung ad coll., provides a review about updated information on this topic.
The review is well developed and approached. First it provides information about RNA viruses and relative immune responses. Then it deepens the different platforms to develop vaccines against RNA viruses, taking into account traditional and innovative ways such as inactivated and live-attenuated vaccines, and more recent and recently studied approaches such as vector vaccines, DNA and mRNA vaccines, and subunit, phage-like particle vaccines and lipid nanoparticles vaccines. Also artificial intelligence (AI) is considered as a tool to predict new antigens and improve vaccine formulation.
In my opinion the review should need a minor revision on the English language in some sentences, such rephrasing, error corrections, or better wording.
Points to be addressed:
Line 24 “real-world application”?
Line 48 variants
Lines 49-53, 56-58 rephrase
Line 85 also il6 and il8 are inflammatory and chemotactic
Lines 98-100 rephrase
Lines 142, 147 what do you mean with “antigenic antibody concentration”?
Line 179 what do you mean with “the T cell pool contrasts”?
Lines 181- 182. Please provide some details of the epigenetic changes you refer to here
Line 187 please write extended name for DNKV
Lines 191-193. More rapid and …
Figure 1 and figure 2. please magnify all the writings as they are too small
Line 217 how does it work formaldehyde inactivation?
Line 228 please provide manufacturing company details of the vaccine. Also when you talk about other vaccines against covid-19 (lines 370-372)
Line 261 add LAV here
Line 269 please provide some details about reverse genetics
Line 276 please don’t use et al. in the text
Subheading 3.2.1 and following. If you think it is relevant, you can change the order of the subheading according to the lettering in figure 2
Line 463 “sate booster”?
Line 489 as you are talking about the potential of AI in vaccine production and optimization, I’d suggest making a specific sub-chapter or provide more details in the parts dealing with this evolving technology
Line 506 provide the year of publication of Sia et al.
Subheading 4. Case studies of RNA virus vaccines
I suggest adding also the vaccine development for Nipah virus (check recent relevant literature)
Lines 562-565 please explain here why the Dengue vaccine “is only recommended for individuals with prior dengue virus infection”
Line 568 and following. Please explain the meaning of “afucosylated antibodies”
Line 589 why did the Zika cases have dramatically declined? Do you refer to the efficacy of a vaccine?
Lines 610-610 please provide some details of the “self-replicating RNA platforms”
Line 617 please specify GISAID
Lines 678 and following. See comment about AI
Line 689 please specify here what an “original antigenic sin” is
Line 726 World Health Organization (WHO)
Author Response
Dear Reviewer,
Thank you for your thoughtful review of our manuscript on vaccine advances against RNA viruses. We appreciate your positive remarks regarding the content and structure of our review, as well as your constructive feedback.
We will carefully address your suggestion for minor revisions related to the English language. We will go through the manuscript to rephrase sentences, correct any errors, and enhance wording for clarity and fluency as follows:
Line 24 “real-world application”?
Response: We have revised the “real-world application” to “practical application” in line 23.
Line 48 variants
Response: Thank you for highlighting this error. We have edited it in line 50.
Lines 49-53, 56-58 rephrase
Response: We appreciate this insightful suggestion. We have rephrased it as follows:
Lines 49-53: “In recent years, the impact of RNA viruses on global health and the economy has been profound. RNA virus outbreaks have caused of widespread morbidity, mortality, and economic disruptions. Since 2019, an unprecedented crisis, the COVID-19 pandemic which was caused by SARS-Cov-2, over 7 million deaths and more than $10 trillion has been lost in economic output.” to “In recent years, RNA viruses have had a substantial impact on both global health and regional and global economies. Outbreaks of these viruses have resulted in widespread morbidity, mortality, and economic disruptions, particularly since the onset of the COVID-19 pandemic in 2019, caused by SARS-CoV-2, which is estimated to have caused over 7 million deaths, along with estimated economic losses exceeding $10 trillion [2,3].” in lines 52 -56.
Lines 58-58: “Vaccines are designed to stimulate the immune responses to recognize and against pathogens, which providing individual and herd/population immunity.” to “…vaccines are designed to stimulate the immune responses of hosts against pathogens, thereby providing both individual and herd immunity…” in lines 60-61.
Line 85 also il6 and il8 are inflammatory and chemotactic
Response: We agreed that there are different kinds of inflammatory cytokines than IL-1 and TNF-alpha, so we have revised this part as follows: “These responses are complemented by the activity of inflammatory cytokines, such as interleukins (ILs) and tumor-necrosis factors, that contribute in recruiting immune cells, including macrophages, natural killer cells, and dendritic cells, to the site of infection.” in lines 89-91.
Lines 98-100 rephrase
Response: We have rephrased the “Adaptive immunity, in which the immune system is antigen-specific, directed against pathogens and retains immune memory, is the main mechanism by which vaccines elicit their protective function.” to “Adaptive immunity, characterized by an antigen-specific response against pathogens and a capacity to retain an immune memory, is the primary mechanism via which vaccines confer their protective effects.” in lines 104-106.
Lines 142, 147 what do you mean with “antigenic antibody concentration”?
Response: We have revised the “antigenic antibody concentration” to “antigen-specific antibody concentration” in both lines 159 and 165.
Line 179 what do you mean with “the T cell pool contrasts”?
Response: Thank you for pointing this out, we’ve streamlined the section of adaptive immunity by removing information from line 177-181 that was relatively less relevant, focusing instead of the key concept.
Lines 181- 182. Please provide some details of the epigenetic changes you refer to here
Response: We have provided some details here in line 178 -183 as follows: “These changes primarily involve modifications, including histone modifications, DNA methylation, and non-coding RNA regulation, that alter gene expression without changing the associated DNA sequences. For example, the occurrence of specific histone marks, such as H3K4me3 (trimethylation of histone H3 at lysine 4) and H3K27ac (histone H3 at lysine 27 acetylation), has been observed to increase in response to an initial pathogen exposure, thereby enhancing the accessibility of genes essential for initiating inflammatory responses [57].” in lines 202-208.
Line 187 please write extended name for DNKV
Response: Thank you for highlighting this typo. We have included both full name and abbreviation of each virus mentioned here as follows: “…dengue virus (DENV), human immunodeficiency virus (HIV)…” in line 214
Lines 191-193. More rapid and …
Response: We have revised “rapider” to “more rapid” in line 219.
Figure 1 and figure 2. please magnify all the writings as they are too small
Response: Thank you for the suggestion. We have magnified all the texts in both figures.
Line 217 how does it work formaldehyde inactivation?
Response: We have added an explanation as follows: “Formaldehyde inactivation is based on the cross-linking viral proteins and nucleic acids, thereby effectively neutralizing the capacity of the virus to replicate, while preserving its structural integrity.” in line 251 – 254.
Line 228 please provide manufacturing company details of the vaccine. Also when you talk about other vaccines against covid-19 (lines 370-372)
Response: We have provided the manufacturing company/research institute details as follows:
Line 237: “…Similarly, Covaxin, a beta-propiolactone inactivated vaccine developed by Bharat Biotech, has shown promising results in clinical trials for the prevention of COVID-19…” in lines 264-266.
Lines 422-425: “…by a collaborative venture between Oxford University and AstraZeneca plc. (ChAdOx1 nCoV-19), Johnson & Johnson (JNJ-78435735, Ad26.COV2.S), Gamaleya Research Institute of Epidemiology and Microbiology (Sputnik V), and CanSino Biologics lnc. (Convidecia),…”
Line 261 add LAV here
Response: We have revised the title to Live-Attenuated Vaccines (LAVs) in line 303.
Line 269 please provide some details about reverse genetics
Response: We have added details about reverse genetics as follows: “…Recent approaches have included genetic engineering techniques, including reverse genetics, designed to introduce targeted mutations and optimize codons, thereby directly attenuating the virus and enhancing vaccine safety and efficacy …” in lines 312-324.
Line 276 please don’t use et al. in the text
Response: We have removed the “et al” in line 297 and lines 319-320.
Subheading 3.2.1 and following. If you think it is relevant, you can change the order of the subheading according to the lettering in figure 2
Response: Thank you for the suggestion, the order of content in the text was arranged based on how well-established each platform is and the timeline of their development. This provides a logical flow that aligns with the progression of research in the field. However, the order in the figure differs slightly to simplify the visual layout and enhance readability. By adjusting the figure’s sequence, we aimed to make complex information easier to interpret without disrupting the content's scientific accuracy.
Line 463 “sate booster”?
Response: We have corrected this typo.
Line 489 as you are talking about the potential of AI in vaccine production and optimization, I’d suggest making a specific sub-chapter or provide more details in the parts dealing with this evolving technology
Response: We have included the potential and current innovation of AI in section 6 from lines 723-745, which provides details and examples.
Line 506 provide the year of publication of Sia et al.
Response: We have provided the year as follows: “In 2022, Sia et al. …” in line 573.
Subheading 4. Case studies of RNA virus vaccines
Response: We have corrected the subheading.
I suggest adding also the vaccine development for Nipah virus (check recent relevant literature)
Response: Thank you for the suggestion; we have added section 4.6 as follows:
“4.6. Nipah Virus: Prospects and Opportunities
Given its capacity for human–human transmission and the associated high mortality rate reaching up to 75%, the Nipah virus is considered to have pandemic potential [195]. Currently, there are no approved vaccines or treatments for Nipah virus infections, and it has accordingly been identified as a high-priority pathogen according to the World Health Organization and Centers for Disease Control and Prevention [186,196,197]. Recent advances in Nipah virus vaccine research have identified several promising candidates, among which is a recombinant vesicular stomatitis virus (rVSV)-based vaccine that has been demonstrated to elicit long-lasting immunity in non-human primates, with 100% protection maintained even a year after vaccination [198]. Additionally, both the ChAdOx1 NiVB vaccine, developed using a viral vector vaccine platform, and mRNA-1215, an mRNA vaccine platform developed by Moderna, are currently undergoing clinical trials. The ChAdOx1 NiVB vaccine has been shown to provide full protection in hamsters and African green monkeys, with robust immune responses and no viral replication [199,200], whereas the mRNA Nipah vaccine has been found to elicit effective antibody responses in pigs, thereby prompting its ongoing evaluation for human use [201].” in lines 687-703.
Lines 562-565 please explain here why the Dengue vaccine “is only recommended for individuals with prior dengue virus infection”
Response: We have added an explanation as follows: “In individuals who have had no previous infection, vaccination could increase the likelihood of severe dengue fever or hospitalization if subsequently contracting a natural infection post-vaccination [186].” and provide a CDC website as a reference in line 638-641. Since, the mechanism is still unclear but the antibody-dependent enhancement could be one of the possible mechanisms.
Line 568 and following. Please explain the meaning of “afucosylated antibodies”
Response: We have explained the meaning of afucosylated antibody where it first appears in this review as follows “…afucosylated IgG1 antibodies (characterized by an absence of fucose residues on the Fc region of the antibody glycan structure) with a high affinity for the FcγIIIa (CD16a) receptor are correlated with enhanced disease and can exacerbate the disease in mice model...” in lines 191-193.
Line 589 why did the Zika cases have dramatically declined? Do you refer to the efficacy of a vaccine?
Response: We agree that this is not clear. So, we have rewritten as follows: “However, a recent marked decline in Zika cases has significantly limited the occurrence of natural infections, thereby making it difficult to conduct clinical trials for the evaluation of vaccine efficacy.” in lines 668-670.
Lines 610-610 please provide some details of the “self-replicating RNA platforms”
Response: We have revised the terminology of the “self-replicating RNA platforms” to “self-amplifying mRNA vaccine platform” in line 710, which has been explained in lines 507-508.
Line 617 please specify GISAID
Response: We have added the full name of GISAID in line 717.
Lines 678 and following. See comment about AI
Response: We acknowledge the importance of AI in the future of vaccine development. While AI has the potential to contribute to nearly every step of the vaccine development process, there are currently limited publications addressing this topic. As such, we have included the application of AI in our section on future directions and emerging strategies in lines 860-882.
Line 689 please specify here what an “original antigenic sin” is
Response: We have revised and added an explanation as follows: “By addressing factors such original antigenic sin, in which initial immune responses limit adaptability to new variants, and potential autoantigen triggers, personalized vaccines will offer more precise and effective protection” in lines 871-873.
Line 726 World Health Organization (WHO)
Response: We have revised the “WHO” to “World Health Organization (WHO)” in line 1066.
Thank you once again for your valuable input. We look forward to making the necessary improvements to our work.
All the best,
Shin-Ru Shih
Research Center for Emerging Viral Infections, Chang Gung University
Department of Medical Biotechnology and Laboratory Science, College of Medicine, Chang Gung University
259 Wen-Hwa 1st Road, Kwei-Shan, Taoyuan 333, Taiwan
Tel: +886-3-2118800 ext. 5497
Fax: +886-3-2118174
Email: srshih@mail.cgu.edu.tw
Reviewer 4 Report
Comments and Suggestions for Authors
Kuei-Ching Hsiung et al. submitted an interesting review about vaccines for RNA viruses. The topic was of a certain significance nowadays, and fell within the scope of Vaccines. The manuscript could be considered for publication after a Minor Revision. Please refer to the following comments.
1. As the type of manuscript was Review, please do not use structured abstract.
2. Between Section 2 and 3, it was recommended to add a section about the formulation design of vaccines.
3. In Section 5, some new technologies for vaccines should be discussed in details, like nanotechnology.
4. Some ongoing clinical trials about RNA virus vaccines could be added in Section 6.
5. There was text with lots of different fonts and typefaces. Please unify the format.
Author Response
Dear Reviewer,
Thank you for your insightful feedback on our review manuscript regarding vaccines for RNA viruses. We appreciate your acknowledgment of the significance of our topic and your suggestions for improvement. Below, we address each of your comments:
Comments 1: As the type of manuscript was Review, please do not use structured abstract.
Response 1: We have confirmed with the editor that a structured abstract is indeed the current requirement for MDPI publications. We will ensure that our abstract adheres to this format.
Comments 2: Between Section 2 and 3, it was recommended to add a section about the formulation design of vaccines.
Response 2: We agree that including a section on vaccine formulation design will enhance the manuscript. We will add this section in the introduction part of Section 3, discussing key considerations and innovations in vaccine formulation, as follows in lines 221-228:
- Vaccine Platforms for RNA Viruses
Humans are continually striving to develop effective strategies for the control of RNA viruses that are responsible for numerous diseases, including the influenza virus, dengue virus (DENV), human immunodeficiency virus (HIV), and SARS-CoV-2 [59]. The rapid rates at which RNA viruses mutate pose a perpetual threat to public health, necessitating the development of innovative and effective vaccine platforms. However, although traditional vaccines have provided a foundation for disease prevention, the ever-evolving nature of RNA viruses necessitates continual advances in modern vaccine technologies that will contribute to providing more rapid and adaptable responses to emerging viral threats, thereby ensuring better protection from and control of outbreaks. The development and implementation of these diverse vaccine strategies will be essential for maintaining global health and combatting future pandemics [60,61]. Whereas vaccine platforms constitute the core mechanisms underlying the generation of immune response, it is the vaccine formulations, which encompass the combination of constituents, including the antigen, adjuvants, stabilizers, and delivery systems, that contribute to optimizing stability, delivery, and efficacy [62]. This is particularly important for the manufacture of RNA virus vaccines, as effective formulations can enhance immune responses, ensure stability against rapid mutation, and improve delivery, ultimately enhancing protection against emerging viral threats.
Comments 3: In Section 5, some new technologies for vaccines should be discussed in details, like nanotechnology.
Response 3: We appreciate your suggestion to elaborate on new technologies, such as nanotechnology. We will expand this section to include a more detailed discussion of nanotechnology and cryo-EM, and their implications for RNA virus vaccines as follows in line 748-758:
Nevertheless, despite these multiple challenges hindering vaccine development, scientific innovation has found, and will continue to find, effective solutions. Addressing these difficulties requires not only adaptation but also breakthroughs in vaccine technology. Recent technological and scientific advances, particularly in nanotechnology and cryo-electron microscopy, have significantly improved RNA viral vaccines. Nanotechnology has facilitated considerable progress in vaccine delivery by enabling precise engineering of nanoparticles, such as LNPs, which are essential for stabilizing mRNA vaccines and facilitating cellular uptake, as demonstrated in COVID-19 vaccines [223,224]. Moreover, nanoparticles can be designed to mimic viral structures, thereby enhancing immune system recognition and response [224]. In addition, by resolving viral protein structures at near-atomic resolution, enabling detailed mapping of essential antigens, such as spike proteins, cryo-electron microscopy, has further advanced vaccine design [225]. This technology not only clarifies complex protein structures but also aids in improving pre-fusion-stabilized proteins, thus optimizing these for immune recognition and enhancing vaccine efficacy against rapidly evolving pathogens [226]. Other developments include ongoing advances in adjuvants and delivery systems that will contribute to boosting vaccine efficacy by enhancing the immune response at smaller doses, thereby facilitating more efficient protection [227]. An example in this field is graphene oxide (GO), a water-soluble single-layer carbon that spontaneously adsorbs different antigens. As an adjuvant, GO recruits antigen-presenting cells and induces both cellular and humoral immune responses [228,229]. Among the next generation of adjuvants, researchers have developed small-molecule immune potentiators as TLR7 agonists and optimized these using aluminum salts (alum) for vaccines. For example, the TLR7-based adjuvant AS37 has been shown to elicit enhanced immunogenicity in animal studies and has now entered clinical evaluation, with results to date indicating a good safety profile and effective immune activation in humans, consistent with earlier findings in non-human primates [230,231]. As novel delivery methods, nasal vaccines against COVID-19 are now in phase 1 trials sponsored by the National Institutes of Health, aiming for higher efficacy and reducing the need for needles, which will predictably contribute to improving compliance with existing vaccines [232,233]. These innovations not only address current issues but also bolster our preparedness for future pandemics.
Comments 4: Some ongoing clinical trials about RNA virus vaccines could be added in Section 6.
Response 4: We have included references to ongoing clinical trials in various sections throughout the review (Line 263-265: Covaxin, Line 458-460: live-attenuated bacterial vector vaccines, Line 523-524: INO-4800 as SARS-CoV-2 DNA vaccine, Line 693-696: Nipah virus vaccines, Line 779: mRNA COVID-flu vaccine), as we believe this provides a more integrated view of the current landscape. Given the dynamic nature and uncertainty surrounding clinical trials, we felt that a summarized section might not capture the most up-to-date information effectively. However, we will ensure that these references are clearly indicated to enhance clarity.
Comments 5: There was text with lots of different fonts and typefaces. Please unify the format.
Response 5: We apologize for the formatting inconsistencies and will ensure that the text is unified in terms of fonts and typefaces throughout the manuscript.
We believe these revisions will enhance the clarity and depth of our review. Thank you once again for your valuable feedback. We look forward to submitting a revised manuscript that addresses your comments.
All the best,
Shin-Ru Shih
Research Center for Emerging Viral Infections, Chang Gung University
Department of Medical Biotechnology and Laboratory Science, College of Medicine, Chang Gung University
259 Wen-Hwa 1st Road, Kwei-Shan, Taoyuan 333, Taiwan
Tel: +886-3-2118800 ext. 5497
Fax: +886-3-2118174
Email: srshih@mail.cgu.edu.tw
Round 2
Reviewer 2 Report
Comments and Suggestions for Authors
Unfortunately, the authors did not correct the structure of their review, only made minor edits that did not address the main shortcomings of the article
In the section on adaptive immune response, a set of scattered facts about the influence of T cells on defense against RNA viruses is presented, and there is no consistent logic in the statement. For example, in the sentence “In cases of dengue, CD8+ T cells have been demonstrated to offer cross-protection against different serotypes [30], whereas secondary influenza infections have been found to be associated with a significantly more rapid increase in T cells in the lungs and an accelerated viral clearance [31].” it appears as if CD8 T cells do not offer cross-protection against influenza viruses. Although it is well known that T cells in principle have broad cross-protectivity due to the fact that they form to more conserved epitopes of viral antigens, mainly to stable internal proteins.
There are some other misleading statements, such as “For example, antibody titers have been shown to be correlated with protection against COVID-19 and the response to influenza vaccinations [45,46]”. But this is obvious that antibody titers are the measure of the response to vaccination, and splitting the sentence into two parts where the first talks about protection from one infection, and the second about the immune response to another (as if there is no correlation with protection) does not make scientific sense and can mislead the reader. Or “Similarly, inactivated vaccines are being assessed for the control of emerging RNA viruses, such as Zika, and have been shown to protect pregnant marmosets from Zika infection [67], RSV, and SARS-CoV-2.” Here, the referred paper was focused on Zika virus, but not RSV and SARS-CoV-2. Another example: “In this regard, given that RNA viruses are mutation-prone, studies have found that the pre-existing immunity prevents live-attenuated influenza vaccines from establishing immune responses, notably T cell responses”. What were the authors trying to say? What does pre-existing immunity causing failure to stimulate T cells with a live vaccine have to do with the susceptibility of the virus to mutations? There are many other statements that do not have clear meaning, and is very difficult to grasp the main idea that the authors are trying to convey.
Figure 1 is also misleading. The figure shows that live and inactivated vaccines undergo the same processing through antigen-presenting cells, although the fundamental difference between these two types of vaccines is that live vaccines replicate in the infected cell, and, firstly, due to this, trigger the innate immunity, which inevitably affects the development of adaptive immunity, and, secondly, the endogenous pathway of antigen presentation in live vaccines is very different from inactivated vaccines, and the cytotoxic response is much stronger for live vaccines than for inactivated vaccines. In addition, the presence of structural antigens in the form of viral particles (but not as individual soluble antigens) also promotes activation of follicular T-helper cells (Tfh), which enhances the B-cell response. This is why a stronger humoral response is formed to vaccines in the form of nanoparticles than to vaccines with similar antigen content but in the form of soluble proteins.
In summary, this manuscript has the very ambitious goal of evaluating traditional and new approaches to vaccination against RNA viruses. However, since RNA viruses are very different in their biological characteristics, each group of viruses has its own peculiarities, it is very difficult to find common patterns for all these viruses and to present an overall picture of existing vaccines and new vaccination approaches. For instance, respiratory viruses have their own special class of the most promising vaccines - with mucosal delivery, as they form a barrier at the port of viral entry. Moreover, even different respiratory viruses have their own characteristics, for example, RSV – which are known by VAERD after formalin-inactivated whole-virion vaccines. Therefore, it is necessary to correct the review and make it more structured so that each statement of the authors is clear and understandable, highlighting the main features/similarities/differences in approaches to vaccination for different classes of RNA viruses. Otherwise, we end up with a chaotic retelling of selected publications, without a systematic analysis of what you need to pay attention to first.
Comments on the Quality of English LanguageThe authors stated that the paper was edited by a professional English editing service, but this seems not true, as some problems are still here (e.g. “In this review examines…” “…chemical agents, such formaldehyde, or by applying heat”, “…bioreactors that are and efficient and conducive”, “…which contributes to the accumulate mutations…” etc.
Author Response
Dear Reviewer,
Thank you for your detailed and thoughtful feedback on our manuscript. We appreciate the time and effort you have taken to provide constructive comments, which are invaluable in improving the quality of our work. Below, we address your concerns point by point and outline the major revisions we have made to address the issues raised.
General Comments on Manuscript Structure
Comment:
"Unfortunately, the authors did not correct the structure of their review, only made minor edits that did not address the main shortcomings of the article."
Response:
We acknowledge your concern regarding the structure of the manuscript. In the revised version, we have reorganized the introduction and the following sections to present a more cohesive narrative, ensuring each section has a logical progression. We have restructured the “Immunity induced by Viral Infections and Vaccinations” section to highlight the roles of innate and adaptive immunity response to viral infections and vaccine-induced immunity. A dedicated subsection now addresses the case studies of RNA virus vaccines, with an introduction about how vaccine development fights against the RNA virus. Moreover, a new summary table clarifies key points and reduces the impression of scattered information.
Section on Adaptive Immune Responses
Comment:
"In the section on adaptive immune response, a set of scattered facts about the influence of T cells on defense against RNA viruses is presented, and there is no consistent logic in the statement…"
Response:
We agree that the section required clarification and a more logical flow. The revised section now begins with an overarching explanation of the importance of immunities gained via natural infection and vaccination-induced immunity. We have revised “In cases of dengue, CD8+ T cells have been demonstrated to offer cross-protection against different serotypes [30], whereas secondary influenza infections…” to “In the case of dengue, CD8+ T cells have been shown to provide cross-protection against multiple serotypes [36], while secondary influenza infections are associated with a notably faster expansion of T cells in the lungs and more rapid viral clearance [37].” in lines 137 to 140.
Clarifications on Vaccine Examples and Statements
We appreciate these observations and have revised the manuscript to improve accuracy and clarity. We have revised the "For example, antibody titers have been shown to be correlated with protection against COVID-19 and the response to influenza vaccinations…” to the “For example, antibody titers correlate strongly with protection against both COVID-19 and influenza, reflecting vaccine efficacy [56,57].” in lines 190-191, the "Similarly, inactivated vaccines are being assessed for the control of emerging RNA viruses…” to the “Similarly, inactivated vaccines are being assessed for the control of emerging RNA viruses, such as Zika, and have been shown to protect pregnant marmosets from Zika infection [75].” in lines 284-286, the "In this regard, given that RNA viruses are mutation-prone…” to the “Given that RNA viruses are mutation-prone, study has shown that pre-existing immunity limits live-attenuated influenza vaccines’ ability to generate or boost T cell responses, particularly by hindering the formation of de novo T cell populations [104].” in lines 347 to 350.
Comment:
"Figure 1 is also misleading. The figure shows that live and inactivated vaccines undergo the same processing through antigen-presenting cells…"
Response:
Thank you for pointing this out. Please find the attached file as revised Figure 1. Figure 1 has been completely revised to accurately reflect the differences in antigen processing and immune activation between live-attenuated and inactivated vaccines. The revised figure now highlights: 1) Replication of live-attenuated vaccines in APCs and other host cells, triggering the endogenous antigen presentation pathway and a robust cytotoxic T cell response. 2) The role of soluble antigens in inactivated vaccines, primarily activating the exogenous pathway and humoral immunity.
Comment:
"It is very difficult to find common patterns for all these viruses and to present an overall picture of existing vaccines and new vaccination approaches…"
Response:
We acknowledge that the diversity of RNA viruses presents significant challenges to developing universal vaccination strategies. This review examines various vaccine development approaches and explores potential technological innovations that could expedite vaccine development during an X-virus pandemic. Additionally, we have included a supplementary table summarizing the key features, similarities, and differences among vaccine platforms for different RNA viruses.
We hope these substantial revisions address your concerns and improve the manuscript's quality and coherence. Thank you for your valuable input, and we look forward to your feedback on the revised version.
All the best,
Shin-Ru Shih
Research Center for Emerging Viral Infections, Chang Gung University
Department of Medical Biotechnology and Laboratory Science, College of Medicine, Chang Gung University
259 Wen-Hwa 1st Road, Kwei-Shan, Taoyuan 333, Taiwan
Tel: +886-3-2118800 ext. 5497
Fax: +886-3-2118174
Email: srshih@mail.cgu.edu.tw

Round 3
Reviewer 2 Report
Comments and Suggestions for Authors
The authors adjusted their manuscript according to the critique raised during peer review. The authors summarized the vast amount of published material on the various vaccine platforms used to create vaccines against various RNA viruses. The content of the review itself is quite good, but the manuscript still needs to be finalized to ensure that the reader understands the general idea behind it.
For example, the abstract requires substantial revision, as some phrases are not scientifically sound, particularly, the Results section. Thus, the sentence “Additionally, ongoing trials for vaccines against influenza, Zika, and dengue demonstrate current advances in the efficacy and the scope of these platforms” sounds inconsistent because it is not clear which platforms they are referring to. Besides, the conclusions are not specific either, too general phrases applying different approaches to combat RNA viruses. The main conclusions of this review need to be more specific.
As for the main text, since the authors overview only a part of all known RNA viruses, it is recommended to add a summary table where the selected viruses are listed with their peculiarities and brief description of vaccines already on the market or specific problems encountered during attempts to develop them. The table which authors provide in Supplementary is very confusing. For each virus, several vaccine platforms are listed in one cell (not all, especially for influenza) and then for all these platforms, key features/approaches, challenges, similarities and differences are listed in one cell each, but clearly each vaccine platform has its own advantages and limitations, and all of this cannot be described in one short phrase. This table does not provide any useful information at all, as it will not be clear to the reader which platform these advantages and disadvantages refer to. This table should be completely redesigned and included in the main body of the paper, rather than in the Supplement, because it should summarize the conclusions that the authors draw when reviewing such a huge body of literature data.
Since the authors make their main focus on the mRNA platform, it is necessary to mention the rare side effects that were associated with mRNA COVID-19 vaccines, which were associated with the molecular mimicry phenomenon (e.g. doi: 10.34172/bi.2023.27494, doi: 10.3390/v15051045). For the AdV-vectored vaccines, serious side effects were also noted, which should be reflected in this review.
Please ensure that the figures are of good quality, as they are fuzzy right now.
Author Response
Dear Reviewer,
We sincerely thank you for your detailed and constructive feedback on our manuscript. Your insights have been invaluable in improving our work's clarity and scientific rigor. Below, we provide a point-by-point response to your comments detailing the revisions made.
Comment:
"The abstract requires substantial revision, as some phrases are not scientifically sound, particularly the Results section. For example, the sentence 'Additionally, ongoing trials for vaccines against influenza, Zika, and dengue demonstrate current advances in the efficacy and the scope of these platforms' sounds inconsistent because it is not clear which platforms they are referring to."
Response:
The abstract has been rewritten to enhance scientific clarity and precision. Specific platforms, such as mRNA vaccines, have been explicitly mentioned where relevant. The result section is as follow: “The case studies underscore the successful application of RNA-based vaccines, particularly in the fight against COVID-19, which has saved millions of lives. Current clinical trials for influenza, Zika, and dengue vaccines continue to show promise, highlighting the growing efficacy and adaptability of these platforms. Furthermore, artificial intelligence is driving improvements in vaccine candidate optimization and providing predictive models for viral evolution, enhancing our ability to respond to future outbreaks.”
Comment:
"The conclusions are not specific either, too general phrases applying different approaches to combat RNA viruses. The main conclusions of this review need to be more specific."
Response:
We have revised the conclusion section as follows:
“Advances in vaccine technology, such as the success of mRNA vaccines against SARS-CoV-2, highlight the potential of nucleic acid platforms in combating RNA viruses. Ongoing trials for influenza, Zika, and dengue demonstrate platform adaptability, while artificial intelligence enhances vaccine design by predicting viral mutations. Integrating these innovations with the One Health approach, which unites human, animal, and environmental health, is essential for strengthening global preparedness against future RNA virus threats.”
Comment:
"Since the authors overview only a part of all known RNA viruses, it is recommended to add a summary table where the selected viruses are listed with their peculiarities and brief descriptions of vaccines already on the market or specific problems encountered during attempts to develop them."
Response:
We rewrote the intro part of the case study and added a new summary table to the main text. This table lists the selected RNA viruses, the peculiarities of each, and a concise description of available vaccines or the specific challenges encountered during vaccine development as follows: “The development of vaccines against RNA viruses has advanced rapidly in recent years. Various vaccine platforms have proven to be effective in combating a wide range of RNA viruses, marking a new era in vaccine development. This section examines case studies of several RNA viruses, highlighting the different vaccine types developed, the challenges encountered during their development, and the technological innovations that have shaped the current landscape of RNA virus vaccines. A comparative overview of selected viruses is presented, highlighting their unique characteristics, current vaccine status, and the specific challenges encountered during vaccine development (Table 1). “ in lines 614-622.
Comment:
"The table which authors provide in Supplementary is very confusing... This table should be completely redesigned and included in the main body of the paper."
Response:
The supplementary table has been redesigned for clarity and included in the main body. Each vaccine platform now has its own row, and the respective advantages, limitations, and challenges are described in separate columns. This revised table has been moved into the main body of the manuscript to better summarize the conclusions, and an extra section was added as follows:
“4.7 Comparison of Vaccine Platforms Against Selected RNA Viruses
To provide a clearer comparison of vaccine platforms, we created a table highlighting the different platforms used for developing vaccines against selected RNA viruses (Table 2). This table presents an overview of the traditional and modern vaccine approaches, including inactivated, live-attenuated, mRNA, DNA, and viral vector vaccines, and their applications in combating viruses such as SARS-CoV-2, influenza, Zika, and dengue. It serves as a concise reference to understand the strengths and limitations of each platform in the context of these viral threats.” in lines 751-757.
Comment:
"Since the authors make their main focus on the mRNA platform, it is necessary to mention the rare side effects that were associated with mRNA COVID-19 vaccines, which were associated with the molecular mimicry phenomenon. For the AdV-vectored vaccines, serious side effects were also noted, which should be reflected in this review."
Response:
We have expanded the discussion of mRNA and AdV-vectored vaccines to include rare side effects, referencing the suggested studies. This information has been integrated into the end of the SARS-CoV-2 case study section to provide a balanced overview of the benefits and risks of these platforms. As follows: “Vaccine-induced side effects, particularly with adenoviral vector-based vaccines, have raised concerns. Severe thromboembolic events such as cerebral venous sinus thrombosis (CVST) and splanchnic vein thrombosis (SVT), often accompanied by thrombocytopenia, have been reported 4–14 days after vaccination. No such events have been observed with mRNA-based vaccines. This has led to the identification of vaccine-induced immune thrombotic thrombocytopenia, likely caused by an immune-based response to adenoviral vectors [190,191]. In contrast, mRNA vaccines, which have significantly reduced COVID-19 morbidity and mortality, have been associated with rare cardiovascular complications such as myocarditis, acute coronary syndrome, thrombosis, and hypertension. These events may be linked to angiotensin-converting enzyme 2 (ACE2) dysregulation caused by spike protein mimicry. Despite these rare risks, the benefit-risk ratio of mRNA vaccines remains highly favorable, with medical monitoring recommended for individuals with pre-existing cardiovascular conditions [192].” in lines 650-662.
Comment:
"Please ensure that the figures are of good quality, as they are fuzzy right now."
Response:
We have provided editors with higher-resolution versions to ensure clarity and readability, and we will ensure that the final proof version maintains high-quality standards.
We hope that these revisions address your concerns adequately and improve the quality of our manuscript. Thank you once again for your time and effort in reviewing our work. We look forward to your further feedback.
All the best,
Shin-Ru Shih
Research Center for Emerging Viral Infections, Chang Gung University
Department of Medical Biotechnology and Laboratory Science, College of Medicine, Chang Gung University
259 Wen-Hwa 1st Road, Kwei-Shan, Taoyuan 333, Taiwan
Tel: +886-3-2118800 ext. 5497
Fax: +886-3-2118174
Email: srshih@mail.cgu.edu.tw

Round 4
Reviewer 2 Report
Comments and Suggestions for Authors
The authors revised their manuscript according to the comments made. In general, the review is suitable for publication, but it should be ensured that the table(s) is(are) included in the body of the paper. Now in the manuscript the authors refer to two tables, but in fact there are no tables in the body of the paper, and only one table is attached as a supplement.